# Imputing single-cell protein abundance in multiplex tissue imaging

Raphael Kirchgaessner [1,2], Cameron Watson [1,2], Allison Creason [1,2], Kaya Keutler [3] & Jeremy Goecks [1,4] ✉

Multiplex tissue imaging enables single-cell spatial proteomics and transcriptomics but remains limited by incomplete molecular profiling, tissue loss, and probe failure. Here, we apply machine learning to impute single-cell protein abundance using multiplex tissue imaging data from a breast cancer cohort. We evaluate regularized linear regression, gradient-boosted trees, and deep learning autoencoders, incorporating spatial context to enhance imputation accuracy. Our models achieve mean absolute errors between 0.05–0.3 on a [0,1] scale, closely approximating ground truth values. Using imputed data, we classify single cells as pre- or post-treatment, demonstrating their biological relevance. These findings establish the feasibility of imputing missing protein abundance, highlight the advantages of spatial information, and support machine learning as a powerful tool for improving single-cell tissue imaging.

Multiplex tissue imaging (MTI) are a set of single-cell spatial proteomics and transcriptomics assays for highly detailed profiling of biological tissues. With MTI, single-cell abundance levels and spatial distribution of 10–150 proteins and/or 500–2000 RNAs can be quantified simultaneously[1,2], MTI enables characterization of individual cells as well as tissue organization, and MTI has been used in studies of healthy tissue[3], COVID[4], cancer[5], and other diseases[6–8], There are many MTI platforms, including cyclic immunofluorescence (CycIF)[9], CO-Detection by indEXing (CODEX)[10], CosMx[11], Xenium[12] and multiplex immunohistochemistry[13]. MTI has been used to generate large datasets in NIH consortia such as the NIH Human BioMolecular Atlas Program[14] and the NCI Cancer Moonshot Human Tumor Atlas Network[15]. MTI is also an increasingly common assay in cancer[16], where it has proven important for quantifying tumor spatial organization and microenvironment heterogeneity[17] and connecting these features to cancer subtypes, prognosis, and therapy response[18,19].

However, several key factors limit the usefulness of MTI. Only 10–150 proteins and/or several thousand RNAs can be assayed in a single experiment, and hence the information obtained from a single experiment is bounded. Further, MTI assays can suffer from several technical issues that reduce the information obtained, including tissue loss or folding, probe failure, illumination artifacts, or errors in downstream image processing. These limitations greatly impact MTI data quality and substantially reduce the overall utility of MTI. To mitigate these limitations and improve utility of MTI, machine learning and deep learning approaches can be used to computationally increase the numbers of proteins/RNAs available from MTI and mitigate assay failures. Computationally increasing−or imputing−additional data by filling in missing data with predicted values is already common in other molecular assays, such as single-cell RNA sequencing (scRNA)[20–27], bulk genomics[28] and bulk transcriptomics[29]. While imputation has been applied to MTI images[30,31], to the best of our knowledge imputation on MTI single-cell datasets has not been explored. Imputation has been applied to MTI image data (refs. 32,33, being able to reconstruct protein expression in images. However, imputing single-cell data is especially valuable because single-cell datasets require fewer computational resources to process than images and can be readily integrated with other molecular datasets.

In this study, we applied machine learning (ML) and deep learning (DL) methods to impute protein abundance in tissue-based cyclic immunofluorescence (t-CyCIF)[9] datasets obtained from breast cancer tissues. Because t-CyCIF is an open and quantitative multiplexed tissue imaging assay, it is ideally suited for imputation. We evaluated the performance of ML and DL methods to predict protein abundance

[1]Department of Biomedical Engineering, Oregon Health & Science University, Portland, OR, USA. [2]The Knight Cancer Institute, Oregon Health & Science University, Portland, OR, USA. [3]Department of Chemical Physiology and Biochemistry, Oregon Health & Science University, Portland, OR, USA. [4]Department of Machine Learning, Moffitt Cancer Center, Tampa, FL, USA. ✉e-mail: jeremy.goecks@moffitt.org

levels in t-CyCIF single-cell datasets that included 20 proteins. Three distinct ML/DL approaches—regularized linear regression, gradient-boosted trees, and autoencoders—were used to impute single-cell protein abundance values across both patients and timepoints. Spatial information was introduced to improve imputation results. To demonstrate a biological application of imputed single-cell protein abundance, we used imputed data to predict whether single cells were more likely to come from pre-treatment or post-treatment breast cancer biopsies. Overall, our results demonstrate that accurate imputation is possible for many proteins, that spatial information significantly improves imputation results, and that imputed protein values are useful in a biological application.

## Results

### Study cohort and analysis overview

The multiplexed tissue imaging single-cell datasets used in this study were generated using a 20-plex t-CyCIF[9] assay applied to a cohort of

hormone receptor-positive (HR +), HER-2 negative metastatic breast cancer biopsies. t-CyCIF is a unique multiplexed tissue imaging assay that has been shown to provide robust and repeatable quantifications of protein concentrations across a range of biological samples. The tissue biopsies and datasets are part of the NCI Cancer Moonshot Human Tumor Atlas Network[15] and have detailed associated clinical metadata. Our dataset includes a total of eight biopsies derived from four patients (Fig. 1a) that received a CDK4/6 inhibitor in combination with endocrine therapy, which is a common combination therapy in metastatic HR+ breast cancer. Each patient contributed a pair of biopsies, a pre-treatment biopsy and a biopsy taken at the time of tumor progression.

Image stacks collected from t-CyCIF were processed using the MCMICRO image analysis pipeline[34] to generate single-cell feature tables (Fig. 1b). Each row in the table is a single cell identified in the image, and the table columns are the protein abundance levels calculated via mean pixel intensity per cell. In total, 475,359 single cells

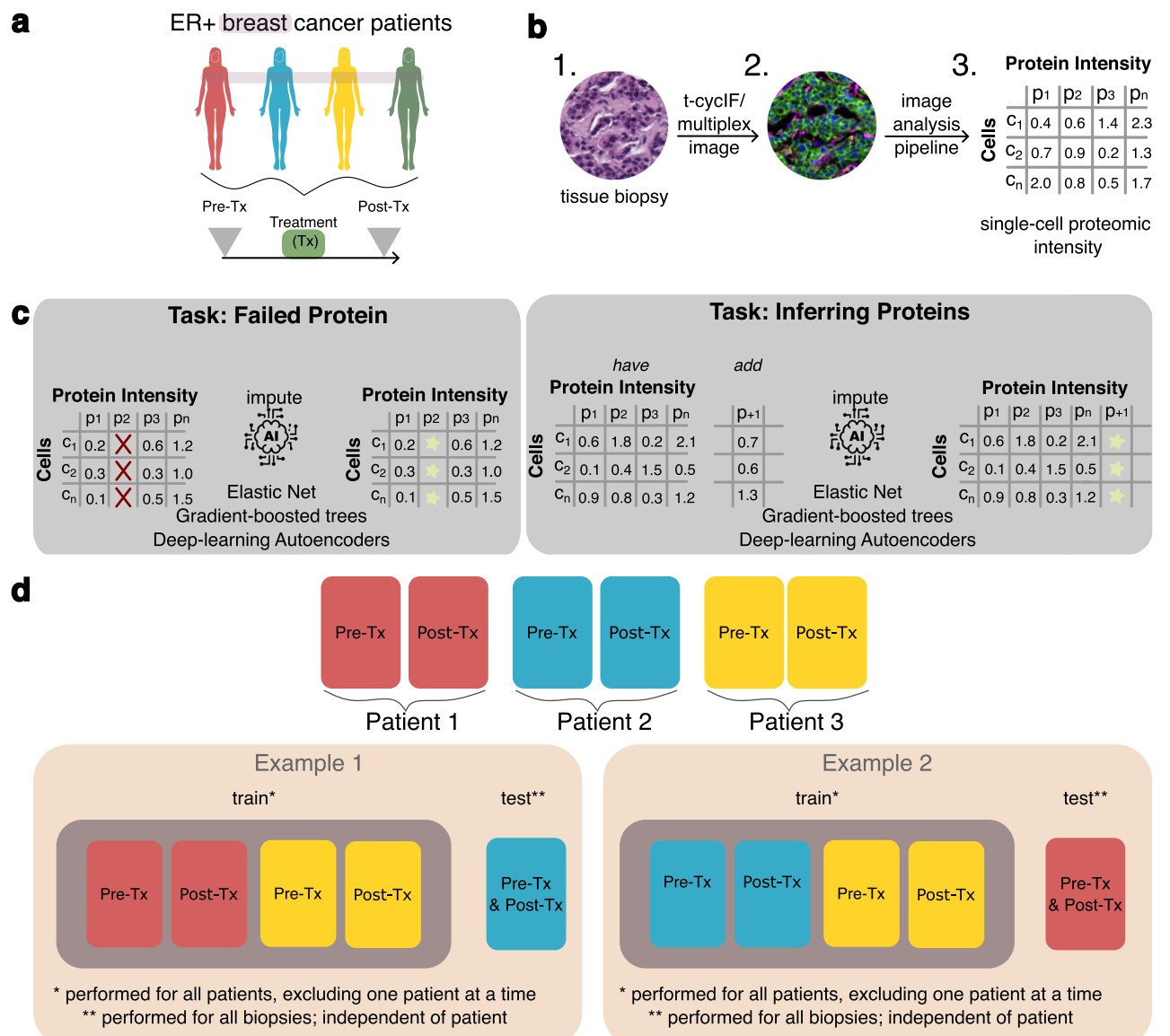

**Fig. 1 | Overview of dataset, study motivations, and analysis approaches. a** Biopsies were obtained from four HR+ breast cancer patients before and after the same therapy for a total of eight biopsies. **b** Each biopsy was assayed using the multiplexed tissue imaging assay t-CyCIF to quantify abundance levels of 20 proteins and then processed using an image analysis pipeline to create single-cell

feature tables (total number of cells identified: 475359); **c** The key tasks addressed by this work are imputing failed proteins and inferring additional proteins not present in an multiplex tissue imaging (MTI) experiment; **d** Approaches for training and testing ML models for imputing proteins across patients.

**Table 1 | Overview of proteins assayed using t-CyCIF and their use as functional or lineage proteins**

| Lineage Protein | Functional Protein |
| --- | --- |
| CD45 | Ki67 |
| αSMA | pERK |
| eCadherin | PR |
| CK19 | EGFR |
| CK14 | p21 |
| CK17 | pRB |
| CK7 | AR |
| Vimentin | HER2 |

Lineage proteins are used to identify cell types whereas functional proteins are used to characterize cell function.

were identified across all biopsies, with an average of 59,400 cells per biopsy. To perform in-patient evaluation, either the pre-treatment or the post-treatment biopsy was used for training a machine learning model while the remaining biopsies were used for testing model performance. Biopsy timing was not used in this study. In total, 16 proteins were shared between all biopsies, including eight proteins for identifying cell types (lineage proteins) and eight proteins for characterizing cellular functional states (functional proteins) (Table 1).

The imputation task in this study was to predict protein abundance levels for a withheld protein or set of proteins. Preprocessing was performed to remove all columns from the datasets except protein intensities, followed by a min-max scaling approach, which maps values between zero and one. Thus, model error is in the range [0,1] where lower error represents better performance. For each machine learning experiment, one or more proteins were withheld and used as the target variable(s) for the predictive model, and the remaining protein abundances were used as input features for the model. This task simulates the key application for imputation in MTI: computationally inferring proteins not originally included in an MTI assay or inferring protein levels where the assay failed (Fig. 1c). Three machine learning methods were used for imputation: elastic-net (EN) regularized linear regression[35], light gradient-boosting machine (LGBM)[36], and neural network autoencoders (AE)[37].

These algorithms offer different advantages for addressing the complexities of our dataset and research objectives. Elastic Net (EN) is a linear model that effectively handles high-dimensional data like that in our MTI datasets by using regularization to manage multicollinearity and predict protein levels. However, EN requires a separate model to predict each protein, and this is time-consuming. Light Gradient Boosting Machines (LGBM) use a non-linear approach for developing predictive models and are amongst the most efficient and best performing methods for tabular data like our MTI datasets. Like EN, LGBM also requires a model for each protein. Autoencoders (AEs) can learn non-linear relationships, reduce dimensionality, and denoise data, allowing a single model to impute multiple proteins at once, although their compression techniques may lead to some loss of precision. Autoencoders were chosen for this study due to their ability to reduce dimensionality and denoise data while preserving essential information. This helps in data imputation and improving data quality before further analysis. EN and LGBM are straightforward to implement and handle linear and non-linear relationships, respectively, while AEs provide efficient preprocessing and multi-protein imputation.

Imputation model training and evaluation were conducted using a leave-one-out cross-validation (LOOCV) approach (Fig. 1d). In this methodology, each patient was considered a single data point, whereby a model was trained on all biopsies except those from one patient. Subsequently, the model's performance was assessed using the biopsies from the patient excluded during training. This LOOCV

approach was chosen to prevent data leakage from biopsies associated with the same patient as the test biopsy, thereby closely approximating real-world application scenarios. Model performance was calculated by averaging the mean absolute error (MAE) scores across all runs of the model on a particular train-test dataset split. Statistical evaluations were carried out using the Mann-Whitney U test, and multiple hypothesis tests were adjusted using the Benjamini-Hochberg correction method.

### Protein abundance imputation with elastic net and light gradient-boosting machines

To establish baseline performance of our imputation models, we conducted a test using mean imputation, where values were imputed by using the mean protein abundance value in the training dataset. Using mean values for imputation serves as a null or baseline model to determine if a machine learning model provides genuine improvements over a simple heuristic. The EN model outperformed the null model by an average of 0.078 MAE indicating that the EN model demonstrated superior performance compared to the null model (Fig. 2a). This performance difference was statistically significant, with an average adjusted p-value less than 0.0001 for all proteins. Proteins CK17 and Ki67 were most accurately imputed with MAE of 0.05. Proteins for which the imputation MAE exceeded 0.2 included CK19, ER, CK14, and PR.

Using Light Gradient Boosting Machine (LGBM) yielded improved imputation accuracy compared to EN (Fig. 2b). Like the EN, LGBM performance for the same 12 of 16 proteins was between 0.05 and 0.20 MAE. LGBM imputation accuracies for CK19 and ER are like the EN and greater than 0.2 MAE. Overall, LGBM displayed more accurate imputation results than EN (Table 2) both in terms of mean and standard deviation. To provide a comprehensive overview of performance, a *mean of all proteins* column is included to show the average imputation accuracy across all proteins for each model (Fig. 2a, b). While using LGBM improved imputation accuracy compared to the EN, some proteins still exhibit a high MAE, such as CK19 and ER. These proteins exhibited high variance (Supplementary Table 2), presenting a significant challenge for accurate imputation. Supplementary Fig 1 shows protein abundance distributions of selected proteins with especially high or low variance to illustrate why imputation is difficult for proteins such as CK19 and ER that exhibit high variance. To further evaluate imputation performance, we calculated the single-cell level correlation between ground truth and imputed protein expression. Correlation ranged from 0.4 (CK19, PR) and 0.8 (EGFR) with an average of 0.56 indicating moderate to strong alignment of the imputed data with the ground truth data (Supplementary Fig 2, Supplementary Fig 3).

We also evaluated imputation accuracy within patients by modifying the LOOCV approach described above. The modified within-patients LOOCV approach included one biopsy from each patient in the training dataset and used the remaining biopsy from the same patient for testing. Surprisingly, imputation accuracy in the across-patient LOOCV approach was higher than imputation accuracy in the within patients for some proteins (Supplementary Fig. 4). We hypothesize that this performance difference may be attributable to the more diverse training dataset in the across-patient approach. This diverse training dataset may enable imputation models to better handle heterogeneity across patients.

We further assessed imputation performance using additional metrics. Side-by-side visualization of in situ imaging from the original assay, the ground truth single-cell protein abundance values calculated via image processing, and the imputed data show the same tissue structural patterns in all three modalities (Fig. 2c, d, Supplementary Fig. 5). This visual alignment of tissue structure demonstrates that the imputed data preserves the biological structure present in the raw images and the ground truth single-cell data. The adjusted rand index (ARI) and silhouette scores were also calculated using ground-truth and imputed single-cell protein expression values. ARI measures

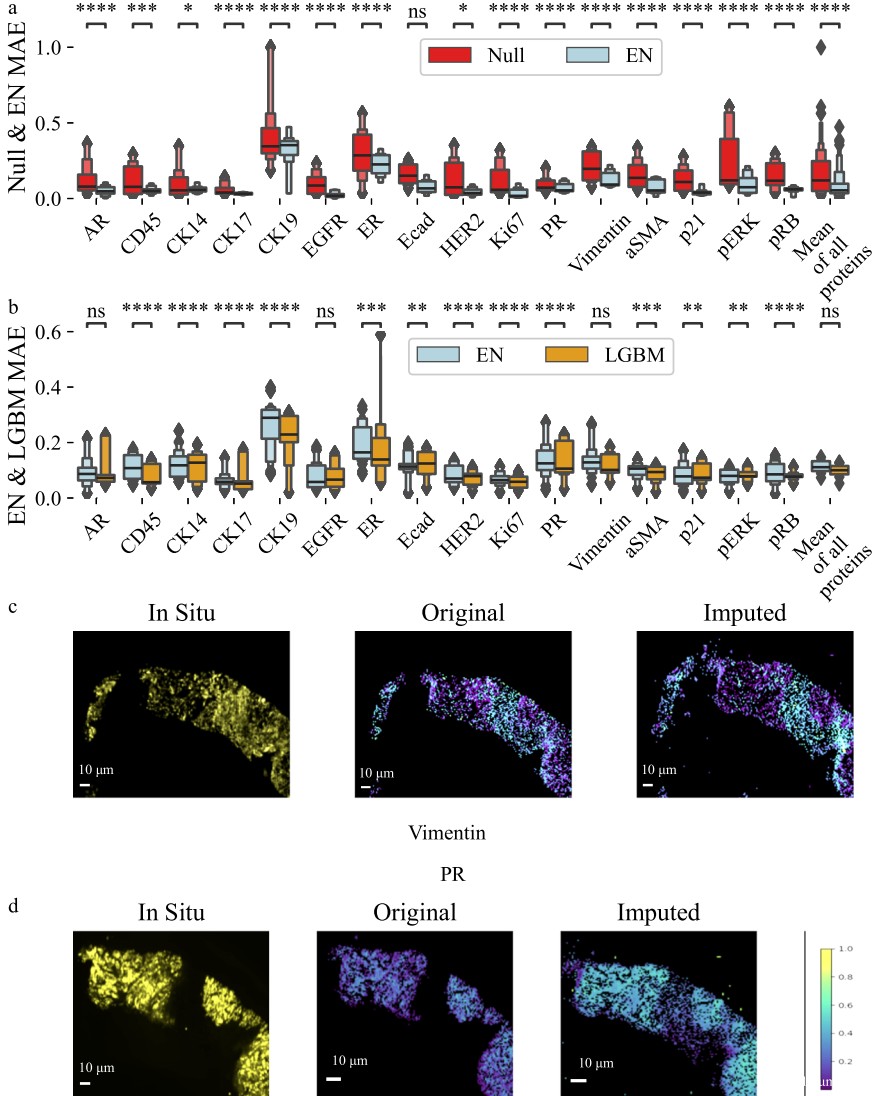

**Fig. 2 | Imputation results for null model and elastic net and Light GBM machine learning models across patients. a** Imputation results for all proteins demonstrate improved mean absolute error (MAE) by using Elastic Net (EN) compared to a null model. **b** Imputation results using EN & Light GBM (LGBM) show low MAE for imputation for 12 out of 16 available proteins. **c, d** Visualization of in situ protein expression, ground-truth single-cell abundance from the image processing pipeline, and imputed single-cell abundance for proteins Vimentin and PR. Results were created using $n = 475359$ single cells. We used 30 replicates with different train & test splits to validate performance metrics. Supplementary Table 3 and Supplementary Table 4 provide all boxen plots description for this figure. *p*-values were calculated using a two-sided Mann-Whitney test and the Benjamini-Hochberg procedure for multiple testing comparisons. Each boxenplot displays nested boxes corresponding to progressively smaller quantile ranges. The central, widest box represents the interquartile range (25th–75th percentiles), capturing the middle 50% of the data. Narrower boxes above and below reflect increasingly extreme quantiles (e.g., 12.5th–87.5th, 6.25th–93.75th), providing a detailed view of distribution tails. Outliers beyond the outermost quantile range are shown as diamonds. p-values: ns: not significant $p \le 1.00e + 00$; *: $1.00e\text{-}02 < p \le 5.00e\text{-}02$; **: $1.00e\text{-}03 < p \le 1.00e\text{-}02$; ***: $1.00e\text{-}04 < p \le 1.00e\text{-}03$; ****: $p \le 1.00e\text{-}04$.

## Table 2 | Mean and standard deviation of performance for EN, LGBM and AE

| Model | EN | LGBM | AE Single | AE Multi |
|---|---|---|---|---|
| Mean (Std) | 0.11 (0.07) | 0.10 (0.06) | 0.13 (0.09) | 0.13 (0.09) |

agreement between two clustering results, with a minimum of zero for no clustering agreement and one for perfect agreement. ARI values between ground truth and imputed data range from 0.6 to 0.77 with an average of 0.69, indicating strong cluster agreement between ground truth and imputed protein expression values (Fig. 3a). The silhouette score measures cluster quality by assessing how tightly data points in a cluster are grouped. Silhouette scores of ground truth and imputed data show that the cluster quality improved by an average of 1.13%

using the imputed data. The silhouette score for imputed data decreased for only CK19 (Fig. 3b).

We also performed a single-cell phenotype analysis using the ground truth and imputed data. For each dataset, we assigned phenotypes using the MCMICRO tool suite[34]. ARI between assigned phenotypes from ground truth data and assigned phenotypes from imputed data averaged 0.72, indicating strong to very strong overlap of phenotypes (Fig. 3c). Further, we assessed how well an LGBM model can assign phenotypes using either ground truth protein expression data or ground truth data replaced with imputed values for one protein. The model was trained and evaluated using the same across-patient cross validation approach as our imputation models. To assess the consistency of phenotype predictions based on imputed data, we computed the Jaccard score between predicted phenotypes derived

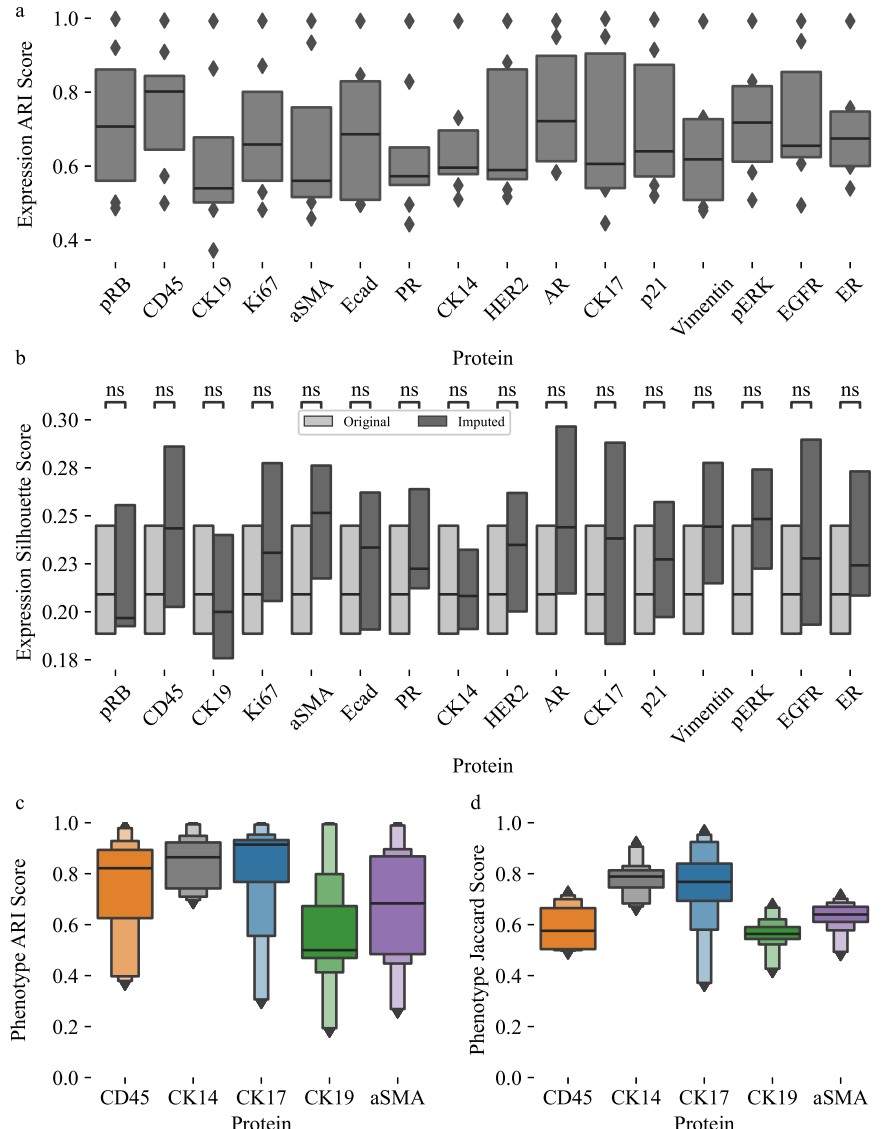

**Fig. 3 | Cluster metrics and Phenotype calling results between original and imputed values. a** Adjusted Rand Index (ARI) scores demonstrating high similarity between ground truth and imputed data clustering. **b** Silhouette scores for single-cell protein abundance clustering improve when using imputed values, indicating improved clustering when using imputed data. **c** Adjusted Rand Index (ARI) Scores for cell phenotype matching using ground truth and imputed data showing moderate to strong overlap. **d** Jaccard scores show moderate to strong overlap between phenotypes using ground truth and imputed protein expression data. Results were created using $n = 475359$ single cells. We used 30 replicates with different train & test splits to validate performance metrics. Supplementary Table 5, Supplementary Table 6, Supplementary Table 7 and Supplementary Table 8 provide detailed descriptions for all boxenplots. *p*-values were calculated using a two-sided Mann-Whitney test and the Benjamini-Hochberg procedure for multiple testing comparisons. Each boxenplot displays nested boxes corresponding to progressively smaller quantile ranges. The central, widest box represents the interquartile range (25th–75th percentiles), capturing the middle 50% of the data. Narrower boxes above and below reflect increasingly extreme quantiles (e.g., 12.5th–87.5th, 6.25th–93.75th), providing a detailed view of distribution tails. Outliers beyond the outermost quantile range are shown as diamonds. p-values: ns: not significant, $p \leq 1.00e + 00$; *: $1.00e\text{-}02 < p \leq 5.00e\text{-}02$; **: $1.00e\text{-}03 < p \leq 1.00e\text{-}02$; ***: $1.00e\text{-}04 < p \leq 1.00e\text{-}03$; ****: $p \leq 1.00e\text{-}04$.

from ground truth data and those obtained using imputed protein values. Since only a subset of proteins in our panel were used for phenotype predictions, the analysis was restricted to these proteins. Jaccard scores ranged from 0.5 to 0.8, with a mean of 0.66, indicating moderate to strong agreement between ground truth- and imputation-based phenotype predictions (Fig. 3d). The Adjusted Mutual Info Score (AMI) as well as silhouette scores (Supplementary Fig. 6) were computed for ground truth and predicted phenotypes as well. AMI scores average 0.73 across all proteins, demonstrating high overlap between ground truth and imputed phenotypes. Silhouette scores from ground truth phenotypes range from 0.25 and 0.3, while scores from imputed phenotypes are comparable but somewhat lower, with a range of 0.2 to 0.28 (Supplementary Fig. 6).

## Protein abundance imputation using autoencoders and all model comparisons

An autoencoder (AE) is a deep learning neural network for accurate reconstruction of high-dimensional data that include two distinct components: (1) an encoder network that maps a high-dimensional input to a lower-dimensional representation in a latent space and (2) a decoder network that reconstructs the original high-dimensional input from the low-dimensional latent space representation. The goal of an AE is to perform information-preserving dimensionality reduction of its input to the latent space so that it can then accurately reconstruct the input from the latent representation. AEs have been successfully used for imputation in various biological domains, including single-cell RNA[25,38–40], genomics[28] and more[41]. Unlike LGBM and EN models, AEs

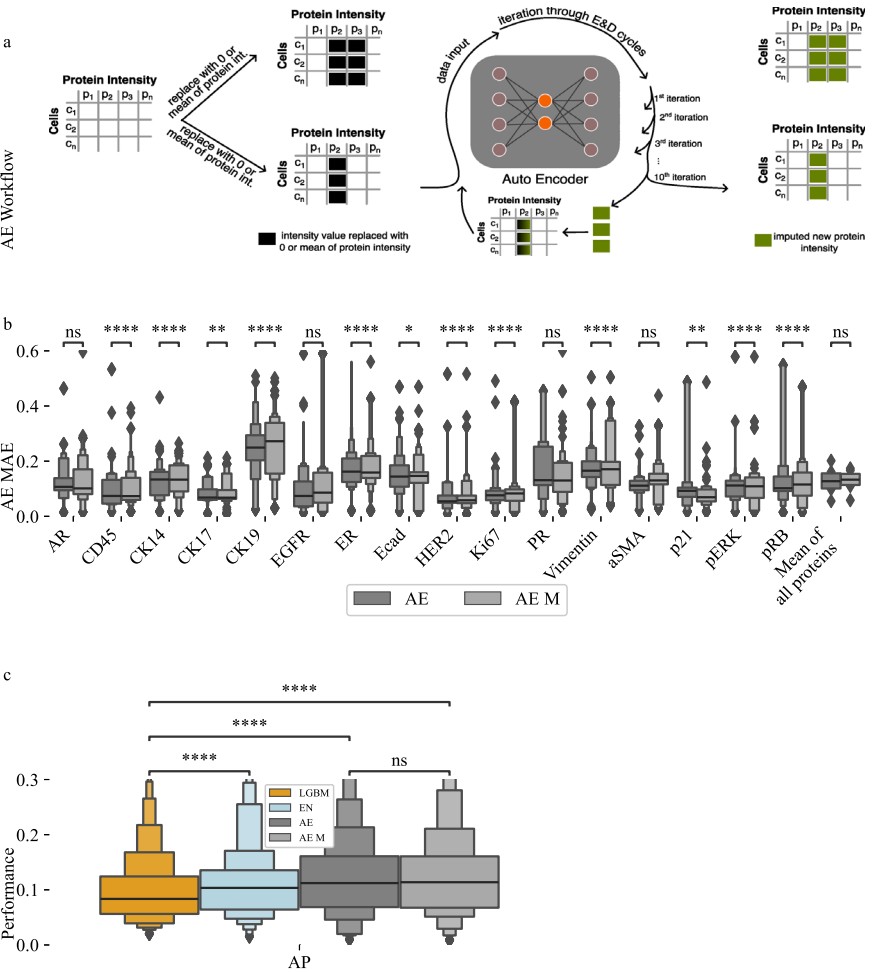

**Fig. 4 | Autoencoder imputation results and performance comparison between machine learning models. a** The autoencoder (AE) is trained and then uses an iterative approach to impute single or multiple proteins. To start, proteins to be imputed are replaced with either zero or the mean of the intensity values in the training set. Then, the autoencoder is used iteratively to predict protein intensities using output values as new input values for each iteration. **b** AE single- and multi-protein imputation performance. **c** performance comparison between all evaluated machine learning (ML) models shows similar performance overall and that Light Gradient Boosting Machine (LGBM) performs best, followed by Elastic Net (EN) and finally AE. There is no significant difference between single and multi-protein imputation performance for AE. Results were created using $n = 475359$ single cells. We used 30 replicates with different train & test splits to validate performance metrics. Supplementary Table 9 and Supplementary Table 10 provide detailed description for all boxplots. $p$-values were calculated using a two-sided Mann-Whitney test and the Benjamini-Hochberg procedure for multiple testing comparisons. Each boxenplot displays nested boxes corresponding to progressively smaller quantile ranges. The central, widest box represents the interquartile range (25th–75th percentiles), capturing the middle 50% of the data. Narrower boxes above and below reflect increasingly extreme quantiles (e.g., 12.5th–87.5th, 6.25th–93.75th), providing a detailed view of distribution tails. Outliers beyond the outermost quantile range are shown as diamonds. p-values: ns: not significant $p \le 1.00e + 00$; *: $1.00e-02 < p \le 5.00e-02$; **: $1.00e-03 < p \le 1.00e-02$; ***: $1.00e-04 < p \le 1.00e-03$; ****: $p \le 1.00e-04$.

can impute multiple features simultaneously due to their ability to fully reconstruct the entire input data. Leveraging this capability, we conducted both single-protein and multi-protein imputation experiments based on the order of protein assays during t-CyCIF's multiple imaging rounds. T-CyCIF involves multiple rounds to stain, incubate, and capture images. Proteins were sequentially removed from each round, and the AE was trained and evaluated for each set of proteins. Initially, proteins from the first round were removed, and the AE was trained and evaluated. This process was then repeated for all proteins in the second round, and so on. To maintain simplicity, no other pairings of proteins were made beyond the rounds.

The AE was trained using biopsies from three patients, including both pre- and post-treatment samples, with biopsies from a fourth patient reserved for validation. Aggregating all biopsy data allowed the AE to develop an internal representation focused on minimizing reconstruction error. During the imputation phase, we initially replaced the target protein's values with the mean expression levels across the dataset. The modified dataset was processed through the

AE, which performed continuous cycles of encoding and decoding to iteratively refine the imputed values. For each cycle, the AE replaced the ground truth protein values with the decoded data from the previous cycle. This iterative process was repeated 10 times, as each protein required a different number of optimal iterations for accurate imputation. We used the mean expression from iterations five to ten as the final imputed value (Fig. 4a).

AEs accurately imputed proteins in both single- and multi-protein experiments (Fig. 4b). Imputation accuracy of CK19 levels is between 0.15-0.35 MAE, while imputation of the best performing proteins, CK17 and p21, is between 0.05 and 0.10 MAE. Like the EN and LGBM models, imputation performance is worst for the proteins with the most variable abundance levels in our breast cancer cohort, including CK19, ER, and PR. We next compared performance for all three machine learning models used for imputation. Overall, LGBM performed best, followed by the EN and the AEs. These performance differences are consistent between models (Fig. 4c). However, performance differences between the models are relatively modest, with the LGBM achieving a mean

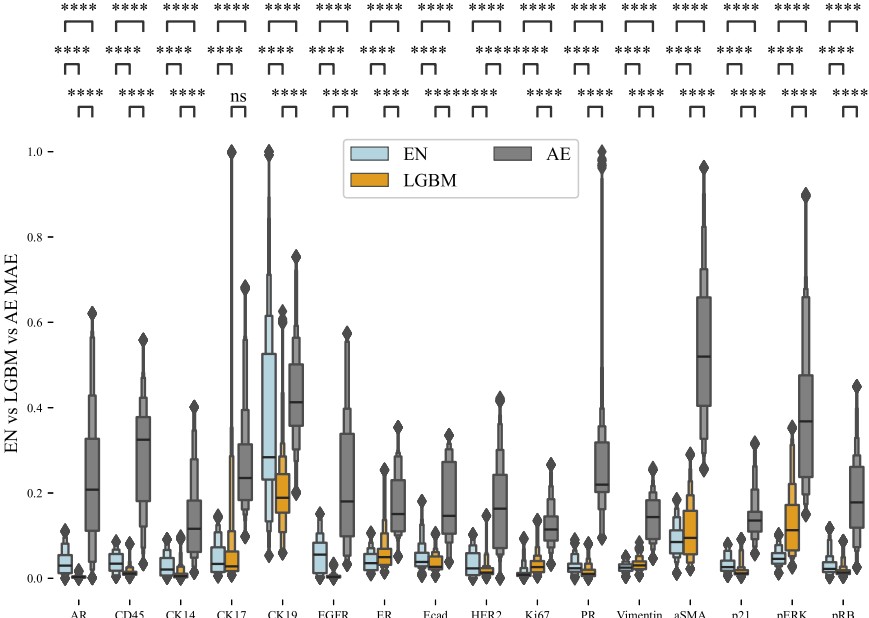

**Fig. 5 | Imputation performance of EN, LGBM, and AE machine learning models on an independent t-CyCIF dataset.** Dataset was obtained from a breast cancer tissue microarray that includes two cores each from 26 tumors. Imputation results are similar to those obtained in our primary cohort and dataset, showing that our imputation methods are applicable beyond the primary cohort to other cohorts and datasets. Results were created using n = 475359 single cells. We used 30 replicates with different train & test splits to validate performance metrics. Table 11 provides a detail overview for all boxenplots. p-values were calculated using a two-sided Mann-Whitney test and the Benjamini-Hochberg procedure for multiple testing comparisons. Each boxenplot displays nested boxes corresponding to progressively smaller quantile ranges. The central, widest box represents the interquartile range (25th–75th percentiles), capturing the middle 50% of the data. Narrower boxes above and below reflect increasingly extreme quantiles (e.g., 12.5th–87.5th, 6.25th–93.75th), providing a detailed view of distribution tails. Outliers beyond the outermost quantile range are shown as diamonds. p-values: ns: $p \leq 1.00e + 00$; *: $1.00e\text{-}02 < p \leq 5.00e\text{-}02$; **: $1.00e\text{-}03 < p \leq 1.00e\text{-}02$; ***: $1.00e\text{-}04 < p \leq 1.00e\text{-}03$; ****: $p \leq 1.00e\text{-}0$.

accuracy of 0.10 MAE, followed by the EN with a mean accuracy of 0.11 MAE, and the AEs with a mean accuracy of 0.13 MAE (Fig. 4c). Auto-encoder performance differences between the imputation of single proteins and that of multi-proteins are minimal.

To further evaluate the performance of imputation in MTI using machine learning, we performed imputation on an additional t-CyCIF dataset from the Human Tumor Atlas Network[15]. This dataset was taken from a breast cancer tissue microarray and included two tissue cores from each of 26 breast cancer tumors. On average, each core included approximately 9850 cells. Unlike the cancers in our main analysis cohort, these cancers are primary disease rather than metastatic and represent all different subtypes of breast cancer. This data is publicly available at the NCI Human Tumor Atlas Portal (see Data Availability section), and the same primary image processing analysis pipeline used in our main analysis was used to generate a single-cell dataset for imputation. From this single-cell dataset of a breast cancer tissue microarray, we extracted the proteins shared with the primary breast cancer dataset discussed previously. We then replicated our imputation experiments using the EN, LGBM, and AE models, employing the same LOOCV approach as before. Results from this analysis show accurate imputation from EN and LGBM models with approximately the same level of overall accuracy as in our main dataset. Accuracy of the AE models was significantly lower in this dataset as compared to accuracy in our main dataset, but it is unclear why this drop in performance occurred. The LGBM model outperformed all other models, while the EN performed better than the AE but worse than the LGBM (Fig. 5, Supplementary Table 2).

**Using cellular spatial information to improve imputation**

A key advantage of MTI datasets is that the spatial coordinates of each cell are known, making it possible to quantify spatial information around individual cells. We hypothesized that the spatial information available in t-CyCIF could be used to improve imputation performance. To test this hypothesis, we quantified the spatial cellular context surrounding a target cell (cell of interest) by calculating the mean protein abundance of neighboring cells. Average abundance levels of all proteins in neighboring cells were then added to our prior set of input features to create a feature set that includes both single-cell protein abundances plus average neighbor abundances (Fig. 6a).

When no neighboring cells were detected, a value of zero was assigned for neighbors' protein abundances. Radii of 15, 30, 60, 90 and 120 micrometers (µm) were used to identify neighboring cells and assess the impact of using different sizes of radii on imputation performance. Only features for one radius setting were used for training a model, and hence a single set of spatial features was included as input for a predictive model.

A radius of 15 µm captures most of the immediate neighbors of a cell, whereas larger radii capture the extended neighborhood of a cell.

We evaluated imputation accuracy using added spatial information in only LGBM and AEs because LGBM performed better than EN and AEs can perform multi-protein imputation. Using spatial information improved overall imputation accuracy for LGBM (Fig. 6b, Supplementary Fig 7), single protein AE (Fig. 7a, Fig. Supplementary Fig 8) and multi-protein AE (Fig. 7b, Supplementary Fig 9). Importantly, imputation accuracy for proteins that had proven difficult to impute well due to their very high levels of variance (see Supplementary Fig 2) was improved significantly with spatial information (Fig. 7a). In particular, imputation of CK19, ER, and PR was much more accurate with spatial information. LGBM performance also improved for other proteins such as CK17, CD45, Ecad, ASMA and p21. Performance of the AE achieved improvements in single-protein imputation for most proteins, with CK19 showing the greatest improvement (Fig. 6a). Multi-protein imputation also benefited from spatial information integration (Fig. 7b). However, performance gains were not as pronounced

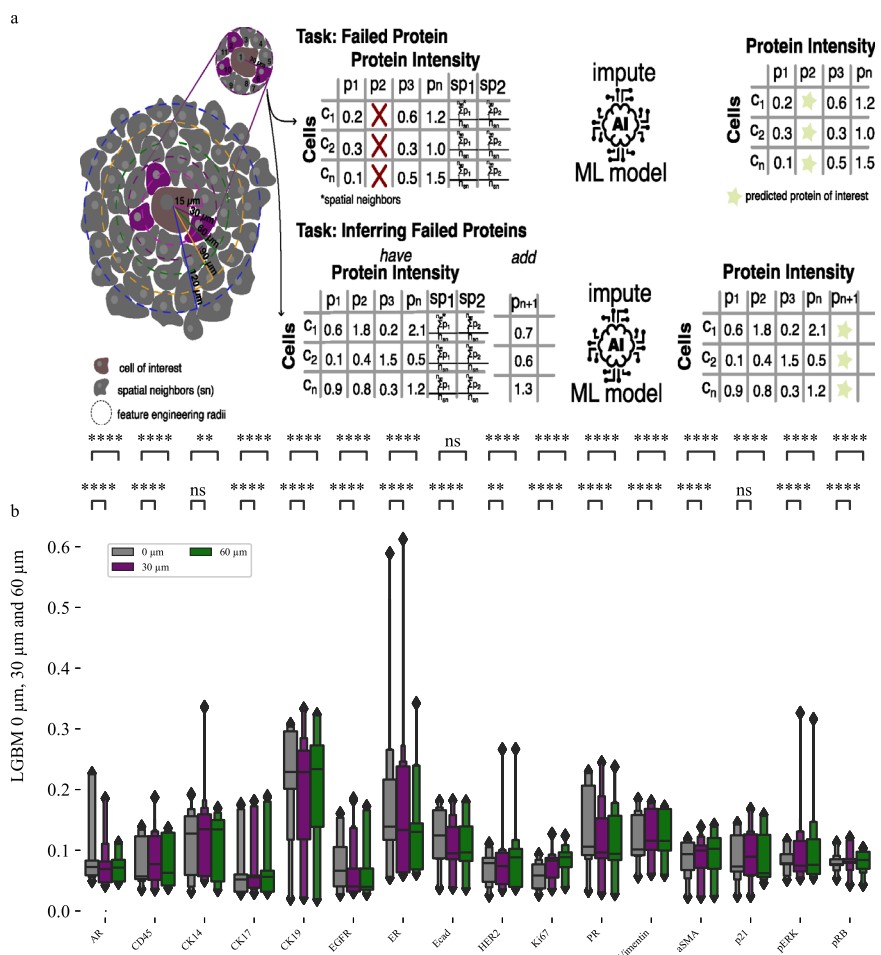

**Fig. 6 | Using spatial information improves imputation performance for LGBM.**
**a** Schematic for creating a feature table based on spatial neighbors found in selected radii. Exemplary 15 μm radius is shown. Red marks the cell of interest (or origin) and protein abundance levels of cells in its neighborhood are averaged to get neighborhood abundance levels. **b** Light gradient boosting machine (LGBM) imputation results across patients with mean absolute error (MAE) scores for 0 μm, 30 μm, 60 μm reveal significant improvement for several proteins such as EGFR, ER, ECAD and PR. Results were created using *n* = 475359 single cells. We used 30 replicates with different train & test splits to validate performance metrics. Supplementary Table 12 provides a detail overview for all boxplots.

*p*-values were calculated using a two-sided Mann-Whitney test and the Benjamini-Hochberg procedure for multiple testing comparisons. Each boxenplot displays nested boxes corresponding to progressively smaller quantile ranges. The central, widest box represents the interquartile range (25th–75th percentiles), capturing the middle 50% of the data. Narrower boxes above and below reflect increasingly extreme quantiles (e.g., 12.5th–87.5th, 6.25th–93.75th), providing a detailed view of distribution tails. Outliers beyond the outermost quantile range are shown as diamonds. *p*-values: ns: $p \leq 1.00e + 00$; *: $1.00e\text{-}02 < p \leq 5.00e\text{-}02$; **: $1.00e\text{-}03 < p \leq 1.00e\text{-}02$; ***: $1.00e\text{-}04 < p \leq 1.00e\text{-}03$; ****: $p \leq 1.00e\text{-}04$.

---

compared to the single protein imputation model. Aligned with prior research[42], imputation accuracy generally improves up to a certain neighborhood radius and then plateaus or declines (Table 3, Supplementary Fig 7, Supplementary Fig 8, Supplementary Fig 9). However, the LGBM does not show the same improvement up until a certain radius, but instead remains largely steady, with a peak performance observed using 60 μm (Table 3, Supplementary Fig 7). Imputation performance is improved by incorporating spatial information (Supplementary Table 1).

**Using imputation to predict treatment timepoints of breast cancer cells**
To evaluate the utility of imputed single-cell protein values from our machine learning models, we used these imputed values to predict whether cells were in pre-treatment or post-treatment timepoints. Using a machine learning classifier that predicts whether single cells are most likely to come from a pre- or post-treatment, we compared classifier accuracy using three different training datasets: (1) ground truth data; (2) ground truth data with a protein's values removed; and (3) ground truth data with a protein's values imputed. The dataset used

for this analysis comprised our primary cohort, consisting of four pre-treatment and four post-treatment biopsies. For modeling, we selected the non-spatial LightGBM (LGBM) model. While the spatial variant demonstrated superior performance overall, it required substantial computational resources. To balance performance and efficiency, the non-spatial LGBM model was used for this analysis.

Initially poor classifier performance was observed because not all biopsy tissues exhibited strong signals associated with a treatment timepoint. To address this issue, we developed a two-step process using two machine learning classifiers (Fig. 8a). First, we selected 300 μm x 300 μm tiles for each biopsy that were associated with treatment timepoint by using a tile machine learning classifier (Fig. 8b). The average protein expression of all cells within a tile was used as input to this tile classifier, and the tiles correctly predicted as pre- or post-treatment by the classifier were selected and used for further analysis. Importantly, the protein to be imputed was removed from the input to this classifier. After the tiles were selected, the second step was performed using cells from the selected tiles. In this step, single-cell imputation was used to produce values for a single protein, and the ground truth plus imputed data was used for timepoint prediction using a single-cell classifier. This

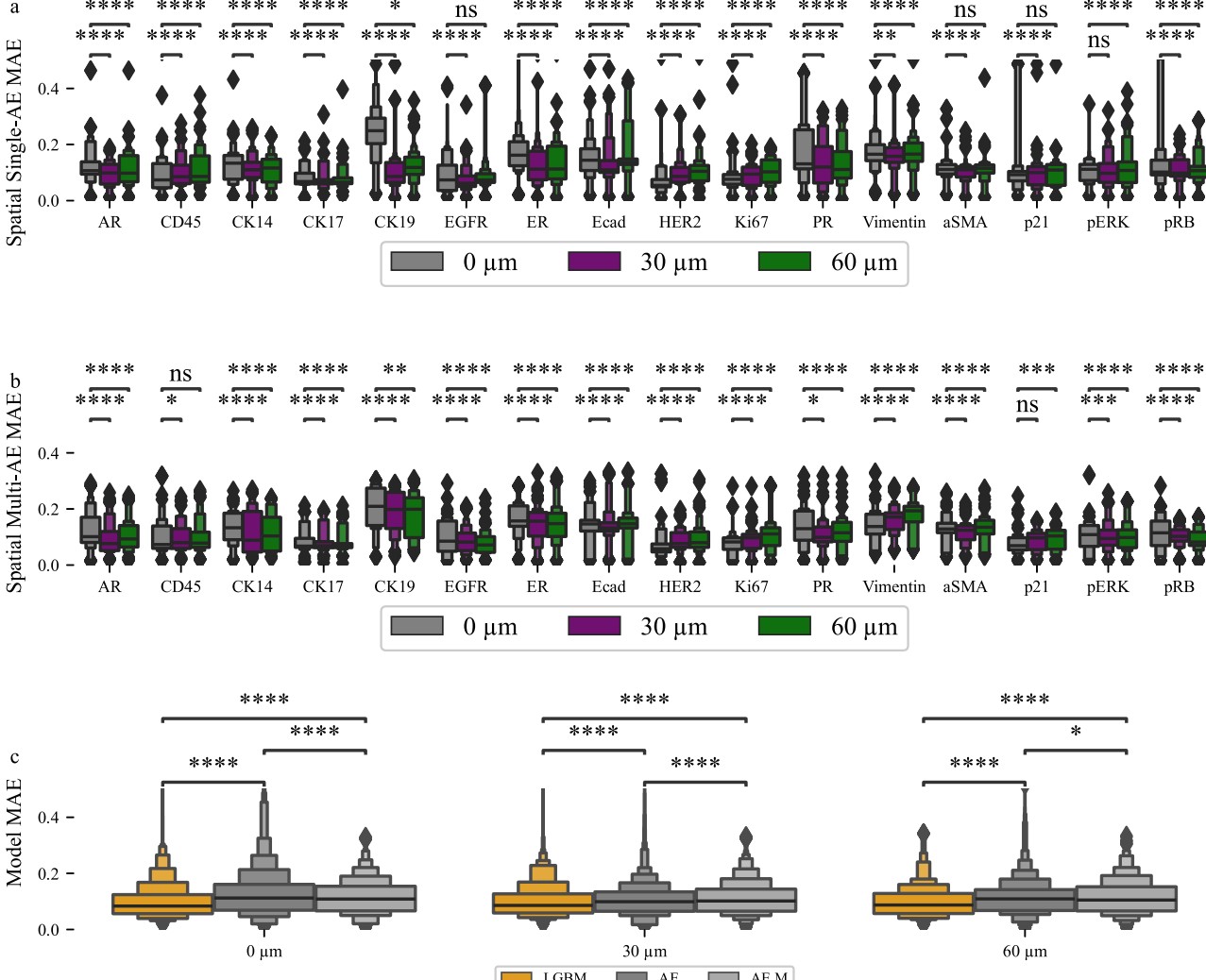

**Fig. 7 | Using spatial information improves imputation performance. a** Single protein imputation mean absolute error (MAE) for 0 μm, 30 μm and 60 μm leads to improved imputation accuracy for proteins such as AR, CK14, CK19, ER and more. Proteins for which imputation improved when using spatial information are in bold and underlined. **b** Multi-protein imputation MAE scores for 0 μm, 30 μm and 60 μm and leads to improved imputation accuracy for proteins such as AR, CK14, CK19, ER and more. **c** Comparison of light gradient boosting machine (LGBM) and autoencoder (AE) imputation performance for 0,30 and 60 μm shows similar performance of all models. Results were created using n = 475359 single cells. We used 30 replicates with different train & test splits to validate performance metrics. Supplementary Table 13, Supplementary Table 14 and Supplementary Table 15 provide detailed boxen plot descriptions for this figure. p-values were calculated using a two-sided Mann-Whitney test and the Benjamini-Hochberg procedure for multiple testing comparisons. Each boxenplot displays nested boxes corresponding to progressively smaller quantile ranges. The central, widest box represents the interquartile range (25th–75th percentiles), capturing the middle 50% of the data. Narrower boxes above and below reflect increasingly extreme quantiles (e.g., 12.5th–87.5th, 6.25th–93.75th), providing a detailed view of distribution tails. Outliers beyond the outermost quantile range are shown as diamonds. p-values: ns: $p \leq 1.00e + 00$; *: $1.00e-02 < p \leq 5.00e-02$; **: $1.00e-03 < p \leq 1.00e-02$; ***: $1.00e-04 < p \leq 1.00e-03$; ****: $p \leq 1.00e-04$.

## Table 3 | Mean and standard deviation of performance for LGBM and AEs for different radii

| Network | 0 μm | 30 μm | 60 μm |
|---|---|---|---|
| LGBM | 0.10 (0.06) | 0.10 (0.06) | 0.10 (0.05) |
| AE Single Protein | 0.13 (0.09) | 0.10 (0.06) | 0.11 (0.06) |
| AE Multi Protein | 0.12 (0.09) | 0.11 (0.06) | 0.12 (0.07) |

Mean and (standard deviation) for each model are listed. 0 μm is considered as baseline without any use of spatial information.

single-cell classifier was trained to classify single cells as either pre-treatment or post-treatment based on protein abundance levels. Accuracy of this single-cell classifier was compared using both imputed values and the ground truth values for each protein.

To prevent data leakage, we employed LOOCV across patients. For each iteration, a biopsy was designated as the test set, while the remaining biopsies, excluding those from the same patient, were used for training the models. We then compared classification accuracy using the ground truth protein abundance data, a control dataset with a protein's values removed, and an imputed dataset where a single protein's expression was replaced with imputed values. Imputation was performed using the LGBM model because it was highly accurate in prior analyses. No spatial data was used in the imputation model for simplicity and so that a comparison between ground truth and imputed data was straightforward.

Overall, classification accuracy from models using imputed data met or exceeded accuracy from models using ground truth data (Fig. 8c). On average classification accuracy was 8.93% higher using imputed data compared to ground truth data. We hypothesize that

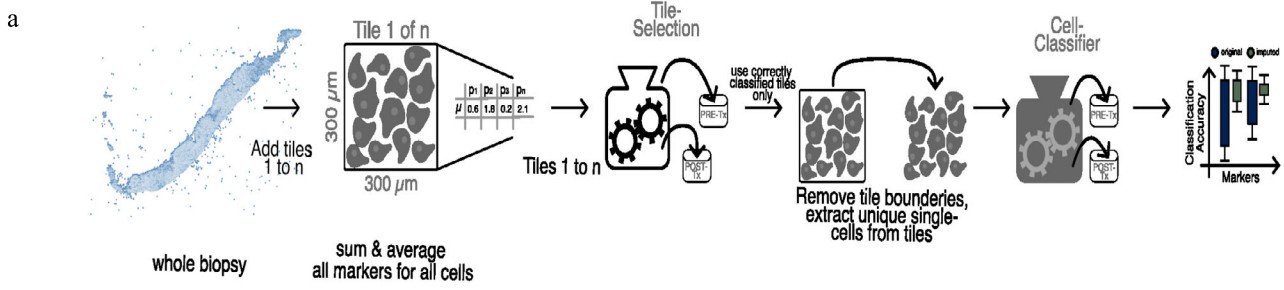

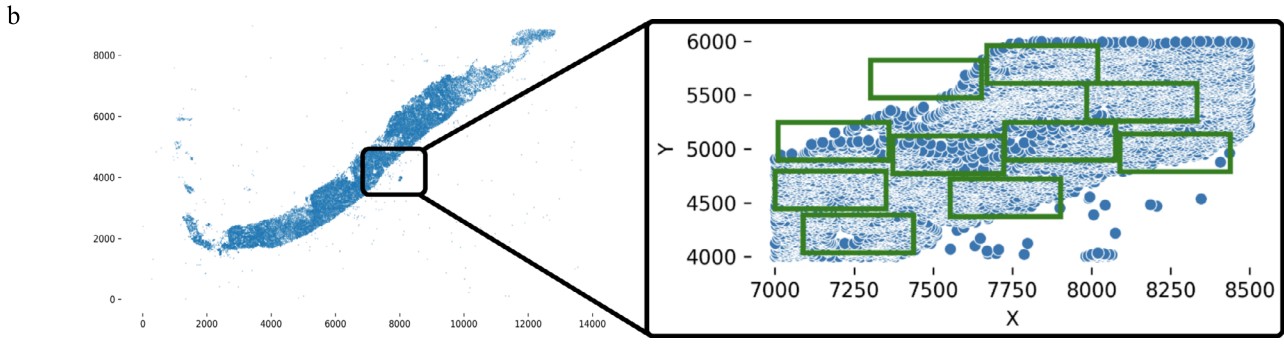

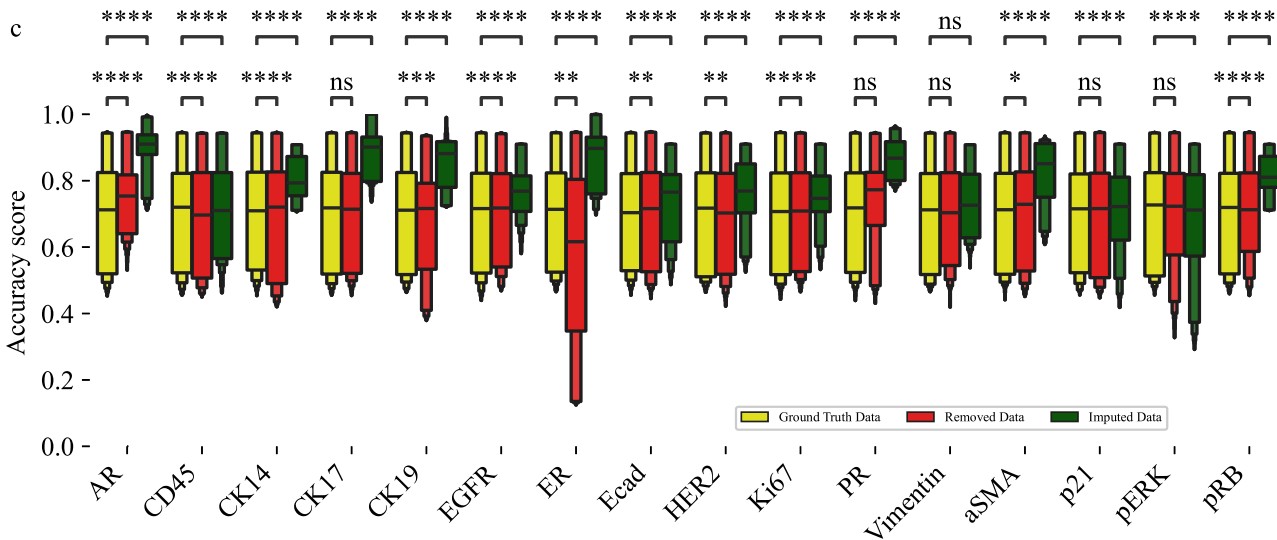

improved classification accuracy using imputed data is the result of noise removal in the imputed data compared to the ground truth data. With less noise, the imputed data led to better performance on the prediction task. Reducing noise not only enhances data quality but also complements the potential effects of upsampling via imputation. Upsampling—amplification of meaningful patterns in the data—has been observed in imputation applications previously[43–45]. Classification accuracy using the control datasets with a protein's values removed were

mixed. For ten proteins, removing their values led to equal or reduced accuracy, but accuracy slightly increased when removing six proteins. When removing a protein increases a model's accuracy, this suggests that the protein is introducing noise to the predictions and possibly harming the classifiers' ability to accurately classify cells. Accuracy of the control models provides additional evidence that the imputation process may be removing noise from the ground truth data, which in turn may help explain higher classification accuracy with imputed data.

**Fig. 8 | Experimental setup and validation for using imputed values to predict treatment timepoints for single cells. a** An initial tile classifier was used to identify tissue strongly associated with treatment timepoints. Next, cells in tissue associated with treatment timepoints were used to train a cell classifier to identify whether cells came from pre-treatment or post-treatment biopsies. **b** Complete biopsy overview, with a zoomed in view depicting green squares which show tiles strongly associated with treatment timepoints. **c** Classification accuracy (higher bar is better) of the cell classifier shows improved performance using imputed values as compared to performance using ground truth or removed protein values. 13125 tiles were used to run the models, with a replicate count of 30.

Supplementary Table 16 provides detailed boxen plot descriptions. *p*-values were calculated using a two-sided Mann-Whitney test and the Benjamini-Hochberg procedure for multiple testing comparisons. Each boxplot displays nested boxes corresponding to progressively smaller quantile ranges. The central, widest box represents the interquartile range (25th–75th percentiles), capturing the middle 50% of the data. Narrower boxes above and below reflect increasingly extreme quantiles (e.g., 12.5th–87.5th, 6.25th–93.75th), providing a detailed view of distribution tails. Outliers beyond the outermost quantile range are shown as diamonds. *p*-values: ns: $p \leq 1.00e + 00$; *: $1.00e\text{-}02 < p \leq 5.00e\text{-}02$; **: $1.00e\text{-}03 < p \leq 1.00e\text{-}02$; ***: $1.00e\text{-}04 < p \leq 1.00e\text{-}03$; ****: $p \leq 1.00e\text{-}04$.

## Discussion

In this study, we utilized machine learning models to accurately impute single-cell protein abundance levels in breast cancer tissue using datasets obtained from the t-CyCIF multiplexed tissue imaging (MTI) assay. Our datasets comprised eight biopsies from a cohort of four metastatic breast cancer patients, facilitating the training and evaluation of these models. Within a range of [0,1], the imputation performance for most proteins exhibited a mean absolute error (MAE) between 0.05 and 0.15. However, proteins with high variance in our cohort, such as CK19 and ER, were more challenging to impute, with MAE ranging from 0.15 to 0.35. Additional evaluation approaches—in-situ visualization, adjusted rand index (ARI), and silhouette scores—also provide evidence that the imputed values are highly comparable to ground truth data. Further, a single-cell phenotyping analysis showed that phenotypes called via imputed and ground truth data are very similar. These results demonstrate that imputed data maintains key information needed for biological applications.

The LGBM model, a gradient-boosted regression tree approach, showed modestly better accuracy than an Elastic Net (EN) model or a deep learning autoencoder (AE). Incorporating spatial features into the models, represented by neighboring cells' protein abundance levels, enhanced their accuracy and reduced the average MAE by 0.02. This improvement was particularly significant for proteins with high variance that were otherwise difficult to impute. This use of spatial information complements recent research indicating that cell communication may vary and requires careful evaluation using multiple cellular neighborhoods[42]. Our results are concordant with this observation as they show a similar pattern of improved protein performance using a diverse set of radii.

While the LGBM shows the overall best performance, there are tradeoffs to consider when choosing a machine learning model for single-cell protein abundance imputation in MTI datasets. Traditional ML models such as LGBM and EN can only impute one protein per model, which requires training and storing a model for each protein to be imputed. Using a single model for each protein is time and cost inefficient. In contrast, an AE can impute multiple proteins at once and even all proteins included in their training data, requiring only a single training session and model. While AEs perform marginally worse than LGBM and EN for protein imputation, their capability for multi-protein imputation offers an advantage in reduced training time and cost. Multi-protein imputation, as opposed to sequential imputation, also models inter-protein relationships, and potentially yields more biologically pertinent relationships to explore.

We have demonstrated robust performance of our imputation methods and biological significance of imputed values. Using an independent MTI dataset from a cohort of 26 breast cancers that included all major subtypes of the disease, our imputation methods showed similar performance to that in our primary cohort. Imputation results across these datasets suggest that our machine learning methods are versatile and can potentially be used in other MTI datasets.

To demonstrate the biological significance of our imputed protein abundance data, classification models were built with the imputed data and used to predict whether individual cells originated from pre-treatment or post-treatment biopsies. Accuracy of these models was high, indicating that single cells can often be classified as associated with a particular disease state. This aligns with previous results that similarly show how single cells can be classified based on disease or treatment state[46–48]. Surprisingly, using imputed data improved classification accuracy compared to using ground truth data. Results from classification models using ground truth data or control data with a protein removed suggest that imputed data may be denoising the ground truth data and leading to improved accuracy. This denoising may reduce errors in the data obtained from the multiplex tissue imaging assay or the primary image analysis workflow used to generate the ground truth data. It is also possible that imputation is enhancing patterns in the ground truth data via upsampling[49,50], and this upsampling via imputation is leading to improved classification accuracy using the imputed data.

Limitations of this work include a focus on protein abundance rather than RNA expression, the small number of proteins used for imputation, the cohort composition of metastatic breast cancers, and the small sample size. These analyses demonstrate that it is possible to impute protein abundance in MTI, but imputation of RNA expression has not been explored. This analysis used the sixteen proteins that were shared amongst all biopsies, and it is uncertain if other proteins can be imputed as accurately as these sixteen. This analysis also focused on breast cancer biopsies and diseased tissue, and imputation results may be different in healthy tissue or in other diseases. A study like ours would benefit from using a larger and more diverse cohort and from different MTI assays with more proteins. Different and larger datasets would help establish the robustness and generalizability of our imputation methods.

In summary, this study demonstrates that machine learning can effectively impute biologically meaningful single-cell protein abundance levels using MTI datasets. Our results provide a foundation for future applications of machine-learning imputed data in single-cell MTI datasets. One potential application is imputation of additional single-cell and cellular neighborhood features, which can in turn aid in understanding tissue ecosystems. Another future application is the use of imputed datasets to predict biomedical outcomes such as tissue response to perturbations or, in the case of disease, response to therapy.

## Methods
### Ethical statement

All biospecimens, data, and consent to use patient samples in research studies were collected under the single-center, observational study Molecular Mechanisms of Tumor Evolution and Resistance to Therapy (IRB#16113). The study was reviewed and approved by the Oregon Health & Science University (OHSU) Institutional Review Board (IRB). All datasets used for training and testing machine learning models are publicly available. Data generation and handling were conducted in accordance with the National Cancer Institute Human Tumor Atlas Network (HTAN) data standards, which are publicly accessible at: https://humantumoratlas.org/standards. All datasets used for training and testing machine learning models are publicly available.

## Experimental Setup

The BOND RX Automated IHC/ISH Stainer was used to bake FFPE slides at 60 °C for 30 min, to dewax the sections using the Bond Dewax solution at 72 °C, and for antigen retrieval using Epitope Retrieval 1 (Leica™) solution at 100 °C for 20 min. Slides underwent multiple cycles of antibody incubation, imaging, and fluorophore inactivation. All antibodies were incubated overnight at 4 °C in the dark. Slides were stained with Hoechst 33342 for 10 min at room temperature in the dark following antibody incubation in every cycle. Coverslips were wet-mounted using 200 μL of 10% Glycerol in PBS prior to imaging. Images were acquired using a 20x objective (0.75 NA) on a CyteFinder slide scanning fluorescence microscope (RareCyte Inc. Seattle WA). Fluor-ophores were inactivated using a 4.5% $H_2O_2$, 24 mM NaOH/PBS solution and an LED light source for 1 h.

The detailed protocol is available in protocols.io (dx.doi.org/10.17504/protocols.io.bjiukkew).

## Data preparation

The source files include X and Y spatial coordinates and bio-morphological information (orientation, area, extent, etc.) for each cell. These features are removed for the initial imputation experiments, which solely rely on protein information.

To prepare the available data for the machine learning models and deep learning networks, we used Min-Max Scaling to scale features to be in the [0,1] range.

## Statistical validity

For robust statistical validity, we conducted more than 30 experiments ($n > 30$) for each protein imputation and each model.

## Elastic net

Elastic Net regression experiments were conducted using the ElasticNetCV implementation from the scikit-learn library, which performs internal cross-validation to select optimal model parameters. Prior to modeling, all data were normalized. Patient-wise leave-one-out cross validation was used. One patient and all data from that patient was placed into a test dataset, and the remaining data from all other patients was used as the training dataset. This approach was done for each patient to create a training and test dataset for each patient that prevented data leakage. The model was trained on the training set and evaluated on the corresponding test set using performance metrics such as mean absolute error (MAE). A separate model was trained for each protein, treating each protein as an individual prediction target.

To ensure statistical robustness and reproducibility, all experiments were repeated more than 30 times ($n > 30$), each with a distinct random seed.

## Light GBM

To set up a training and evaluation pipeline for our Light GBM[36] model, we used the Ludwig[51] platform, which enables "End-to-end machine learning pipelines" in a low code environment. Ludwig models were configured using a YAML-based configuration file specifying the input features—proteins in this study—and the target variable to be imputed. To streamline and scale this process, we automated model setup and execution using a combination of shell scripts and Makefiles. Consistent with all other experiments, train and test sets were generated under a strict constraint: patients selected for evaluation were excluded from the training data and used exclusively for testing. Each run was initialized with a unique random seed to support reproducibility. Performance metrics generated by Ludwig were logged and retained for downstream analysis.

## Autoencoder

An autoencoder-based approach was employed to iteratively impute missing protein values. First, the preprocessed source data were

**Table 4 | Human Tumor Atlas Network (HTAN) biopsy and biospecimen IDs**

| HTAN Biopsy ID | HTAN Biospecimen ID |
|---|---|
| 9 2 1 | HTA9_2_11 |
| 9 2 2 | HTA9_2_21 |
| 9 3 1 | HTA9_3_11 |
| 9 3 2 | HTA9_3_21 |
| 9 14 1 | HTA9_14_6 |
| 9 14 2 | HTA9_14_14 |
| 9 15 1 | HTA9_15_7 |
| 9 15 2 | HTA9_15_15 |

loaded, and train-test splits were generated following the same constraint applied throughout the study: patient data designated for evaluation were excluded from the training set and used solely for testing. To impute a specific protein, its values were initially replaced with the mean of all available entries for that protein.

The prepared dataset was then passed through the autoencoder, which performed a full encoding and decoding cycle (Fig. 2). The resulting output was stored as an intermediate representation for future reference and downstream analysis. From this output, the imputed values for the target protein were extracted and used to replace the initial mean values. This process was repeated for a total of 10 iterations, forming a 10-step iterative imputation pipeline.

Final imputation performance was assessed using the last five iterations: for each patient, the mean imputed value across these five decoding steps was calculated and used to compute the final mean absolute error (MAE) and root mean squared error (RMSE).

To ensure statistical robustness, each model configuration was run a minimum of 30 times, each initialized with a distinct random seed to generate reproducible yet variable results.

## Statistics & reproducibility

Each model was evaluated using cross-fold validation with at least 30 iterations to ensure robust performance estimation. To assess differences in performance between the original and imputed data, we conducted two-sided Mann-Whitney U tests. To account for multiple comparisons, we applied the Benjamini-Hochberg correction. No statistical method was used to predetermine sample size.

No datapoints were excluded from the analyses; the entire available single-cell dataset was used. The models were not trained on the data they were later tested on. Blinding was not needed nor applied during model training and evaluation. Investigators were not blinded to allocation during experiments and outcome assessment.

## Reporting summary

Further information on research design is available in the Nature Portfolio Reporting Summary linked to this article.

## Data availability

The single-cell spatial cyclic immunofluorescence data analyzed in this study and results generated from model predictions have been deposited in the Dataverse database under https://doi.org/10.7910/DVN/RBIJSQ. Table 4 lists the HTAN Biopsy and Biospecimen IDs used in this study. Additional information on these biopsies and biospecimens is available at https://humantumoratlas.org/explore All data associated with the ML experiments and required to generate the plots is available in the Dataverse database under https://doi.org/10.7910/DVN/K0FCQE All TMA data is available through the HTAN Data Portal as part of the HTAN TNP-TMA Project (ttps://data.humantumoratlas.org/).

## Code availability

The source code of this work is freely available in the GitHub repository. https://github.com/goeckslab/MTIProteinImputation.

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

## Acknowledgements

This research was supported by the National Cancer Institute (NCI) of the National Institutes of Health grants U24CA231877, U24CA284167, and U2CCA233280 and by funding from the Prospect Creek Foundation to the OHSU SMMART (Serial Measurement of Molecular and Architectural Responses to Therapy) Program.

## Author contributions

R.K and J.G. both conceived the study. C.W. and A.C. processed the primary image data to produce the single-cell datasets analyzed in this work. R.K. developed the methods, wrote the code, and performed the analysis. C.W., A.C., and J.G. provided feedback and suggestions on methods, code, and analyses. K.K. designed illustrations used in the figures and provided feedback on the overall figure design. All authors read and approved the final paper.

## Competing interests

The authors declare no competing interests.
