## [Transparent Peer Review file · Nature Communications]

Imputing Single-Cell Protein Abundance in Multiplex Tissue Imaging

Corresponding Author: Dr Jeremy Goecks

Version 0:

Reviewer comments:

Reviewer #1

(Remarks to the Author)
Reviewer's Summary

In this manuscript, the authors outline a strategy for imputing single cell protein expression from multiplexed tissue imaging (MTI) data. Missing data in MTI data can occur for several reasons, including technical failure, incompatibility with other affinity reagents in the assay, or simply the lack of space in a particular experiment. Most modern MTI modalities allow between 20-150 markers to be assessed. In this study, the authors preprocess the data down to a [cells] x [markers] table and use this summarized data for model training.

The authors describe three models they adapt for imputation of MTI data: an elastic net regularized linear regression (EN), a light gradient boosting machine (LGBM), and a neural network autoencoder (AE). These are all fairly established regression models with fairly well-known tradeoffs, which the authors note. For the first two, EN and LGBM, one model was trained per marker. For the autoencoder, proteins were (I believe?) iteratively masked during training, and a single model could be used to impute any desired proteins.

The authors do well to test and compare these model frameworks for an important data type, and I appreciate the exhaustive evaluations done. Thought they do not propose a novel method per se, the inclusion of cell neighborhood expression in the imputation is an interesting and useful piece of data. But my excitement for the potential of their approach(es) is dampened by some key problems.

Key Problems:

First, and most importantly, we are left without any sense of the biological meaning of the imputations. In my opinion, the authors really need to use the imputed values in downstream tasks to demonstrate their value. This is spatial data, too – for at least some of the markers, they should show what these imputed values look like in situ alongside the real values.

To provide additional color on the above comment, the authors make several claims about the utility of imputation, but don't provide evidence to back up that utility in their study. If the imputation improves data quality, we should see an impact on a data quality assessment task; if the markers are relevant for clinical predictions, we should see an impact on that task; if the markers are relevant for labeling cell types, we should get more granular or accurate cell labels after imputation. Any of these examples would provide evidence of utility, while mean average error numbers in a vacuum give me very little motivation to use the method. I see that the dataset used has pre- and post- treatment labels, so as a suggestion, predicting that sample level label with and without imputed markers makes a lot of sense as a downstream task.

Second, I have an issue with the training structure. Breast cancer treatment could radically transform the tumor microenvironment, including cell compositions and expression levels of marker proteins. In this study, different treatment conditions were either used to evaluate the model (in-patient) or simply mixed for training and testing (across-patient). This structure exacerbates the first problem above – it draws doubt onto the biological meaning of the imputed proteins - and it significantly decreases my confidence in interpreting mean average error values on their own. To put it another way, if the training data are all over the place, what could the imputed values look like?

Third, there are issues with the presentation of data and conclusions. Specifically:

- How do the authors interpret the across patient performing better vs within patient?
- The authors run significance tests between the across-patient and within-patient groups for every marker; and later for every radius defining cell neighborhood. Several related issues with the statistical tests in the paper as a whole:
 - For IP/AP, these tests set up comparisons that aren't super relevant to the main conclusions.
 - What does it mean to have an average decrease of 0.01 in absolute error over a large population? If this difference is highly significant, was the statistical test useful? Or can the authors give us some intuition about why it's a relevant difference? Maybe it's the wrong test, or the wrong background.
 - In some cases, there are 50 stat tests or more in a single panel. There needs to be correction for multiple testing.
- In Figure 5, "spatial information improves imputation performance for LGBM" – it's not an obvious conclusion. For many markers, error significantly increases with spatial information (Ecad, HER2, Ki67, aSMA...).
- In Figure 6, "autoencoder imputation performance improves significantly using spatial information" – if anything, it's more difficult to see and trust this result given the data shown. And again, abstractly, "significant" is great, but practically speaking, what is the difference in utility between imputation at 0.10 MAE and 0.11 MAE?
- We lack baselines against which to evaluate the MAE values. What does imputing just using the mean value look like in terms of MAE?

Minor Feedback:

- Charts are generally hard to read with the p value stars everywhere.
- I appreciate the clear statement of study limitations!
- The 'impute AI ML Model' graphic is cool, but it would be more useful to the reader to put the names of the models in - in every case.
- Figure 1d – splits used are a little confusing from this graphic.
- Figure 2b – no X axis.
- Figure 2a, 2c – is "Mean" the mean of all imputations? Clarify-
- I know data can be scarce, but some sense of imputation performance, even for just one marker, outside of the training dataset would be really useful to know how the approach generalizes. Another breast cancer dataset?
- What does this mean? "Multi-protein imputation was performed based on the order in which the proteins were assayed during t-CyCIF's multiple imaging rounds".
- In general, explanation of AE training could be clearer. I had to look at the code to understand what was going on.
- I appreciate the inclusion of a clear code repository^
- "to the best of our knowledge imputation on MTI single-cell datasets has not been explored." I have seen a couple related studies, potentially of interest? Wu et al. 2023 [PMID: 37275261] and Pati et al. 2023 [<https://doi.org/10.1101/2023.11.29.568996>]

(Remarks on code availability)

Reviewer #2

(Remarks to the Author)

The manuscript by Kirchgassner, et al evaluate the utility of machine learning methods to improve the validity of multiplexed tissue imaging, where this wet laboratory technique suffers from a number of shortcomings including limited numbers of evaluated protein targets that can be assayed in a single sample, sample integrity issues and suboptimal probe performance. Herein, the authors seek to demonstrate the ability to impute protein abundance at the single cell level using machine learning to augment existing multiplexed datasets, where a breast cancer cohort was used. While there are some promising results, there are significant limitations to the current work as written that should be address, explained as follows:

1. While the machine learning/deep learning (ML/DL) algorithms selected were distinct, it isn't clear how they were selected. Why were these three methods selected? What are the pros and cons to each ML/DL type? How are they representative of the potential ML/DL landscape?
2. There are a variety of acronyms in the figure captions that are not defined therein. They are defined in the main text, but would aid readership if they were redefined in the caption.
3. How was protein abundance defined from the existing dataset? IF and IHC are inherently semi-quantitative methods. Thus, it would be difficult to get an accurate measure of protein abundance for ground truth.
4. How will the ML/DL trained models here extrapolate to other cancer types in other tissues? Will model training be required for each cancer/tissue type? Would the trained ML/DL model perform similarly on metastatic breast cancer specimens?
5. It isn't clear why cytokeratins and hormone receptors exhibited some of the highest variances. The explanation that these proteins are "likely to exhibit patient-specific patterns" is inadequate as it could be argued that this is the case for all stained proteins within the multiplexed imaging dataset.
6. While taking the spatial information into consideration is a strength of this work, it isn't clear how neighboring cells were defined herein? How were the different radii sizes related to number of neighboring cells? It also isn't clear why some cells in the sample would not have neighboring cells?
7. The summary around RNA probes is not relevant to this work and should be removed in its entirety since only protein staining was evaluated.

(Remarks on code availability)

Version 1:

Reviewer comments:

Reviewer #1

(Remarks to the Author)

I thank the authors for addressing many of my concerns, particularly the addition of a patient label prediction task is a good addition. But I maintain that most of the conclusions in the paper are yet supported solely by comparisons of MAE, which is not a biologically meaningful metric. Nor are we given biological intuition for the magnitude of MAE effect sizes, either between model conditions (Null, EN, LGBM, Spatial) or between different markers. This makes the manuscript unidimensional. Other studies cited by this manuscript (scRMD, scImpute) report many dimensions of biologically driven imputation evaluation: MAE yes, but also correlation of RNA concentrations, ARI of cluster results, overlap of differentially expressed genes, cell type identification accuracy, improvement of cell typing, improvement of pathway enrichments etc. The data type of those is transcriptomics, not proteomics, but I believe these are appropriate examples to follow or use as standards.

One easy dimension to add, asked for in review 1 but not done in the revision, would be in situ visualizations of imputed vs ground truth expression in different model contexts, which adds biological intuition to quantitative differences. Cell-wise correlations would likewise give a sense of single cell imputation accuracy and the distribution of marker expression across single cells that is simply not captured by error level.

The addition of the tile-treatment predictions is a step in the right direction, but still a bit messy. The authors used the LGBM model because it was most accurate, but not the spatial LGBM model, which was actually most accurate according to the preceding conclusions. Why? Their result required a fairly involved preprocessing pipeline that includes tile pre-selection. By itself that is OK. But the actual result raises questions as well. First, what exactly is presented? Is it: For each marker, when imputing that marker, the error of predicting treatment status...? Perhaps the Y axis label is incorrect, because increased error would mean worse performance, no? And MAEs close to 1 would be very high in this context? Second, unless I'm mistaken, the tile selection itself uses all the markers, including those markers that will later be imputed, which is an unrealistic scenario. Third, it seems to me that a successful experiment would mean that, in the absence of Marker X, sample level predictions suffer, then predictions can be restored to some extent using imputation. That would be a very strong result.

Perhaps the authors could just do tile classification as in the first step of the current pipeline, but do those with original data, missing data, and imputed data? That would be more interpretable than the cell level classing shown in Fig 7d.

The authors have done well to add treatment-related predictions, some control experiments, and a generalization figure (the second CyCIF dataset). I hope I am being consistent with my original review in saying that it still lacks:

1. Multidimensional evaluation of imputation success
2. A clean experiment showing the impact of imputation on treatment predictions.

(Remarks on code availability)

The repo is documented, clean and usable.

Reviewer #2

(Remarks to the Author)

I have reviewed the changes to the manuscript, which sufficiently satisfy my previous concerns.

(Remarks on code availability)

Version 2:

Reviewer comments:

Reviewer #1

(Remarks to the Author)

I think the authors have addressed my concerns satisfactorily and written good clarifications into the text. Thanks for your replies. I feel a little 'weird' about imputed predictions outperforming ground truth in the prediction task, but accept the

rationale of de-noising as plausible.

The following are only minor points to improve readability or clarity, if the authors or editor should choose:

- * Adjust contrast or thresholds of the in situ / cell expression level images (2c) to make it more visible.
- * Why only 5 proteins for additional metrics (e.g. Jaccard)? Show all, or explain why just a subset
- * Addition of colors with no legend (Fig. 2e, g, h) is slightly confusing.
- * In review doc authors wrote: "the spatial LGBM model requires significant computational resources to train and evaluate." Maybe add this justification of model choice to the paper, or at least mention it when spatial LGBM performs best.
- * The large point size in 7b inset makes the plot confusing, gives the impression points show something more than cell locations.

(Remarks on code availability)

Nice readme!

- We thank the reviewers for their time and effort in evaluating our manuscript. Reviewers identified many strengths of the manuscript as well as several important weaknesses. Reviewers noted the following strengths of the manuscript: (1) imputing single-cell protein abundance is an important and unaddressed problem when using multiplexed tissue imaging; (2) there was a thorough evaluation of different machine learning methods for imputing single-cell protein abundance levels; and (3) use of cell neighborhood expression data improved imputation results. In this revised version of the manuscript, we have made substantial additions and changes to address the weaknesses identified by the reviewers.

Below is a point-by-point response to reviewers' concerns as well as changes made to the manuscript to address these concerns. Reviewers' comments are italicized with wide margins. Our responses are prefixed with an arrow and are indented.

Reviewer #1 (Remarks to the Author):

Reviewer's Summary

In this manuscript, the authors outline a strategy for imputing single cell protein expression from multiplexed tissue imaging (MTI) data. Missing data in MTI data can occur for several reasons, including technical failure, incompatibility with other affinity reagents in the assay, or simply the lack of space in a particular experiment. Most modern MTI modalities allow between 20-150 markers to be assessed. In this study, the authors preprocess the data down to a [cells] x [markers] table and use this summarized data for model training.

The authors describe three models they adapt for imputation of MTI data: an elastic net regularized linear regression (EN), a light gradient boosting machine (LGBM), and a neural network autoencoder (AE). These are all fairly established regression models with fairly well-known tradeoffs, which the authors note. For the first two, EN and LGBM, one model was trained per marker. For the autoencoder, proteins were (I believe?) iteratively masked during training, and a single model could be used to impute any desired proteins.

The authors do well to test and compare these model frameworks for an important data type, and I appreciate the exhaustive evaluations done. Thought they do not propose a novel method per se, the inclusion of cell neighborhood expression in the imputation is an interesting and useful piece of data. But my excitement for the potential of their approach(es) is dampened by some key problems.

Key Problems:

First, and most importantly, we are left without any sense of the biological meaning of the imputations. In my opinion, the authors really need to use the imputed values in downstream tasks to demonstrate their value. This is spatial data, too – for at least some of the markers, they should show what these imputed values look like in situ alongside the real values.

To provide additional color on the above comment, the authors make several claims about the utility of imputation, but don't provide evidence to back up that utility in their study. If the imputation improves data quality, we should see an impact on a data quality assessment task; if the markers are relevant for clinical predictions, we should see an impact on that task; if the markers are relevant for labeling cell types, we should get more granular or accurate cell labels after imputation. Any of these examples would provide evidence of utility, while mean average error numbers in a vacuum give me very little motivation to use the method. I see

that the dataset used has pre- and post- treatment labels, so as a suggestion, predicting that sample level label with and without imputed markers makes a lot of sense as a downstream task.

- Thank you for this great suggestion. We agree that demonstrating the biological utility of imputed values in a downstream task is a crucial enhancement to the manuscript. As recommended for this task, we predicted pre- and post-treatment labels for single cells using both the original and imputed data. Our findings show that, in several instances, prediction accuracy improved with the imputed data compared to the original data. These results suggest that the imputed values are biologically meaningful and valuable for biomedical applications. The downstream task and prediction results are detailed in a newly added subsection of the Results section titled “Using Imputation to Predict Treatment Timepoints of Breast Cancer Cells.” Additionally, we added Figure 7 to illustrate the task setup and visualize results.

Second, I have an issue with the training structure. Breast cancer treatment could radically transform the tumor microenvironment, including cell compositions and expression levels of marker proteins. In this study, different treatment conditions were either used to evaluate the model (in-patient) or simply mixed for training and testing (across-patient). This structure exacerbates the first problem above – it draws doubt onto the biological meaning of the imputed proteins - and it significantly decreases my confidence in interpreting mean average error values on their own. To put it another way, if the training data are all over the place, what could the imputed values look like?

- Thank you for this important question. If there are systematic differences between cell profiles in pre-treatment and post-treatment tumor microenvironments, then we agree that our imputation results would be much less reliable and useful. However, three sets of results from our analyses provide confidence that imputation can be successfully performed across treatment timepoints and that there are no systematic differences between cells in pre-treatment and post-treatment biopsies. These results are:
 1. As discussed in detail below, we compared our imputation results to a simple null model that used mean expression as its only prediction. Our imputation results are substantially better than the null model. Importantly, our imputation results were trained on both pre-treatment and post-treatment data together, indicating that the machine learning models can identify and use patterns across both pre-treatment and post-treatment data together.
 2. In our across-patient analyses, we successfully trained accurate imputation models using a dataset consisting of both pre- and post-treatment biopsies and evaluated them on both pre- and post-treatment biopsies. This is further evidence that single-cell protein expression patterns in pre- and post-treatment biopsies can be combined and used well by a machine learning model.
 3. As discussed above, we completed a downstream task to demonstrate that a machine learning model can differentiate cells most likely to occur in pre-treatment and post-treatment biopsies. As with the models discussed in points 1 and 2, these models were trained on combined pre- and post-treatment datasets.

The consistently accurate results across these lines of evidence provides strong support for our claims that imputation data predicted using machine learning with combined pre- and post-treatment datasets is accurate and biologically meaningful.

Third, there are issues with the presentation of data and conclusions. Specifically:

- *How do the authors interpret the across patient performing better vs within patient?*

- We have discussed this observation ourselves. We hypothesize that the observed trend of across-patient imputation accuracy being better than within-patient accuracy can be attributed to reduced overfitting and improved generalizability achieved through a more diverse training dataset. Accurate model performance across patients suggests that the model is robust and can handle heterogeneity found across patient cohorts. It is important to note that in-patient imputation accuracy is also very good. To ensure that the focus of our analysis is on cross-patient performance—the primary use case of this work—we moved the within-patient performance data to supplementary materials. We have added several sentences in Results section “Study Cohort and Analysis overview” discussing these observations.

• *The authors run significance tests between the across-patient and within-patient groups for every marker; and later for every radius defining cell neighborhood. Several related issues with the statistical tests in the paper as a whole:*

o For IP/AP, these tests set up comparisons that aren't super relevant to the main conclusions.

- Thank you for pointing this out, we have removed those significance tests. As discussed above, we have moved comparisons of within-patient (IP) performance to across-patient (AP) performance to the supplementary materials. This change ensures our manuscript's focus is on its main conclusions.

o What does it mean to have an average decrease of 0.01 in absolute error over a large population? If this difference is highly significant, was the statistical test useful? Or can the authors give us some intuition about why it's a relevant difference? Maybe it's the wrong test, or the wrong background.

- Thank you for your suggestion. We agree that providing context for the statistical significance and practical relevance of the 0.01 decrease in absolute error is crucial. The statistical test applied to detect this difference was indeed significant, indicating that the modest error reduction is unlikely to have occurred by chance, especially in a large dataset. This result suggests a real albeit small difference in performance between the models. The effect size of that decrease—0.01— equates to 1 % given the normalization of all protein expression values to [0,1]. In most instances, this 1 % performance difference will not be noticeable, and any of the models can be used. Autoencoders are a great choice if maximum model performance is not needed because they are sufficiently accurate, and a single model can be used to impute multiple proteins. If maximum accuracy is needed for a particular task, Light Gradient Boosting Machine (LGBM) are the best choice because they are the most accurate. The downside of LGBM is that it is costly to train predictive models for each protein. We have added text to the discussion section to highlight the tradeoffs between model accuracy and the need to train multiple models.

o In some cases, there are 50 stat tests or more in a single panel. There needs to be correction for multiple testing.

- Thank you for pointing this out. We have now applied the Benjamini-Hochberg multiple hypothesis testing correction to our results. Importantly, the application of the Benjamini-Hochberg correction did not alter statistical significance of model performance results, and hence the main conclusions of our manuscript did not change. We also noted the use of this statistical test in the Results subsection titled “Study Cohort and Analysis Overview.”

• *In Figure 5, “spatial information improves imputation performance for LGBM” – it's not an obvious conclusion. For many markers, error significantly increases with spatial information (Ecad, HER2, Ki67, aSMA...).*

• In Figure 6, “autoencoder imputation performance improves significantly using spatial information” – if anything, it’s more difficult to see and trust this result given the data shown. And again, abstractly, “significant” is great, but practically speaking, what is the difference in utility between imputation at 0.10 MAE and 0.11 MAE?

- We have improved our figures to make it clear when spatial information improves imputation results. In Figures 5 and 6, proteins with significant improvements are now highlighted in bold and underlined, making it easier for readers to clearly see the performance enhancements. We agree that the practical difference between 0.10 MAE and 0.11 MAE is small. Our previous comments highlighting the tradeoffs between using many slightly more accurate LGBM models vs. a single slightly less accurate autoencoder apply to these results as well.

• We lack baselines against which to evaluate the MAE values. What does imputing just using the mean value look like in terms of MAE?

- We agree that comparing results to a very simple or null model is appropriate. We have incorporated a new subpanel into Figure 2, which shows the performance of a null model utilizing mean protein values versus our Elastic Net (EN) models. The null model performs significantly worse, with an MAE of 0.078 (7.8 %) higher than that of the EN model, demonstrating that our models offer a meaningful improvement over a simple approach. For enhanced visibility and readability, we employed a bar plot with 95 % confidence intervals, making it easier to discern the performance differences between the null model and the EN model.

Minor Feedback:

• Charts are generally hard to read with the p value stars everywhere.

• I appreciate the clear statement of study limitations!

• The ‘impute AI ML Model’ graphic is cool, but it would be more useful to the reader to put the names of the models in - in every case.

- Thank you for your suggestion. We have adjusted the graph to combine the icons with the model names. This modification should enhance the reader’s understanding of the task and the models involved.

• Figure 1d – splits used are a little confusing from this graphic.

- Thank you for your feedback. We have revised Figure 1d to enhance clarity. Specifically, we removed the IP setting and used the newly available space to provide a clearer depiction of the AP setup. This adjustment improves the explanation of the splits used in our analysis.

• Figure 2b – no X axis.

- Thank you for catching this oversight. We have added an X axis to this figure. Figure 2b presents the expression histogram of a specific marker, with the expression values normalized between 0 and 1. The y-axis represents the number of cells in each expression bin.

• Figure 2a, 2c – is “Mean” the mean of all imputations?

- Yes, the reported MAE in all figures was calculated by averaging the mean absolute error (MAE) scores across all runs of the model on a particular train-test dataset split. We have added a sentence in the Results subsection titled “Study Cohort and Analysis Overview” to better explain this calculation.

Specifically, the bar plots illustrate the aggregated results of all patients across all experimental runs, with each experiment conducted approximately 30 times. Previously, we used boxen (enhanced box) plots to display this data. However, we switched to bar plots with confidence intervals to enhance readability and facilitate easier interpretation of the results. The mean bars represent the average imputation performance over all proteins, while the confidence intervals help convey the variability and reliability of the results more clearly. This change was made to provide a more straightforward and intuitive understanding of the data.

- *I know data can be scarce, but some sense of imputation performance, even for just one marker, outside of the training dataset would be really useful to know how the approach generalizes. Another breast cancer dataset?*
 - We agree that validation of our methods on another dataset is important. To address this issue, we performed imputation on a larger and more diverse t-CyCIF breast cancer dataset. This dataset included 26 tumors from all types of breast cancer. Results of this analysis are displayed in Figure 4 and described in the second Results subsection titled “Protein abundance imputation with elastic net and light gradient-boosting machines”. Analyses from this additional dataset show accurate imputation with about the same level of overall accuracy as in our main dataset.

- *What does this mean? “Multi-protein imputation was performed based on the order in which the proteins were assayed during t-CyCIF’s multiple imaging rounds”.*
 - T-CyCIF utilizes multiple rounds to stain, incubate, and capture images. For performing multi-protein imputation, the Autoencoder was trained by sequentially removing proteins from each round and then evaluating the results. Initially, proteins from the first round were removed, and the Autoencoder was trained and evaluated. This process was then repeated for all proteins in the second round, and so on. To maintain simplicity, no other pairings of proteins were made beyond the rounds.

- *In general, explanation of AE training could be clearer. I had to look at the code to understand what was going on.*
 - Thank you for this suggestion. We have added a paragraph in the AE Results subsection to better explain how the model was trained. This paragraph lists all the steps used to train the AE. Briefly, the Autoencoder was trained using a dataset consisting of biopsies from three patients, including both pre- and post-treatment samples. Biopsies from a fourth patient were reserved for validation as a test set. The training process involved aggregating all available biopsy data from the three patients, allowing the Autoencoder to develop an internal representation. Our training objective focused on minimizing the reconstruction error, enabling the Autoencoder to produce output protein expressions that closely match the input values. During the imputation phase for a specific protein, we initially replaced its values with the mean expression levels across the dataset. This modified dataset was then processed through the Autoencoder, which performed encoding and decoding in a continuous cycle to refine the imputation. This iterative process was repeated ten times to ensure robustness. Since the optimal number of iterations required for peak performance varied among different proteins, we calculated the mean expression from iterations five to ten. This mean was then used as the imputed value for the protein, enhancing the accuracy of our imputation analysis. This strategy accounts for the dynamic nature of protein expression changes during the imputation process and optimizes the overall reliability of the reconstructed data. For each protein and round, the respective proteins were replaced with the mean of the protein expression currently available in the dataset.

- *I appreciate the inclusion of a clear code repository^*
- *“to the best of our knowledge imputation on MTI single-cell datasets has not been explored.” I have seen a couple related studies, potentially of interest? Wu et al. 2023 [PMID: 37275261] and Pati et al. 2023 [https://doi.org/10.1101/2023.11.29.568996]*
> *PMID: 37275261 -> also uses images as input. (See Material & Methods section: Image patch generation)*

- Thank you for bringing those two publications to our attention. While both papers address generation and imputation, they focus exclusively on image data rather than single-cell data. Because different machine learning models and different biological conclusions can be drawn from imaging data and single-cell data derived from images, we believe these publications complement our manuscript. In the Introduction of our manuscript, we now reference these manuscripts and discuss how our analyses are different than those presented in these publications.

Reviewer #2 (Remarks to the Author):

The manuscript by Kirchgaessner, et al evaluate the utility of machine learning methods to improve the validity of multiplexed tissue imaging, where this wet laboratory technique suffers from a number of shortcomings including limited numbers of evaluated protein targets that can be assayed in a single sample, sample integrity issues and suboptimal probe performance. Herein, the authors seek to demonstrate the ability to impute protein abundance at the single cell level using machine learning to augment existing multiplexed datasets, where a breast cancer cohort was used. While there are some promising results, there are significant limitations to the current work as written that should be address, explained as follows:

1. While the machine learning/deep learning (ML/DL) algorithms selected were distinct, it isn't clear how they were selected. Why were these three methods selected? What are the pros and cons to each ML/DL type? How are they representative of the potential ML/DL landscape?

- This is an important question. We have added a paragraph in the first Results subsection to explain how we chose our machine learning models. In brief, we chose our machine learning models to match the data that we're working with. Single-cell proteomics data has many observations (cells) and features (proteins), requiring robust feature selection and regularization. We chose Elastic Net for its combination of L1 and L2 penalties, which handle high-dimensional data by addressing complexity and sparsity, effectively managing multicollinearity, and selecting the most relevant proteins. Light Gradient Boosting Machine (LGBM) was selected for its efficiency in processing large-scale, sparse datasets typical of single-cell proteomics. Its advanced tree-based learning algorithms capture subtle differences and complex patterns within cellular behaviors, making it ideal for analyzing over 500,000 cells. Autoencoders were chosen for preprocessing due to their ability to reduce dimensionality and denoise data while preserving essential information. This helps in data imputation and improving data quality before further analysis. In summary, our selection of Elastic Net, LGBM, and Autoencoders is based on their proven performance and suitability for the specific challenges of single-cell proteomics data. These algorithms, widely used in the ML/DL landscape, offer distinct advantages for addressing the complexities of our dataset and research objectives.

2. There are a variety of acronyms in the figure captions that are not defined therein. They are defined in the main text but would aid readership if they were redefined in the caption.

- Thank you for noting this omission. We updated all figure captions to define acronyms that are used in each figure.

3. How was protein abundance defined from the existing dataset? IF and IHC are inherently semi-quantitative methods. Thus, it would be difficult to get an accurate measure of protein abundance for ground truth.

- Thank you for this excellent question. This work focuses on datasets generated from t-CyCIF, which has been shown to be quantitative by Lin et al. (2018). Titled "Highly multiplexed immunofluorescence imaging of human tissues and tumors using t-CyCIF and conventional optical microscopes", this publication discusses the advancements in cyclic immunofluorescence (t-CyCIF), a technology that enables highly multiplexed, quantitative imaging of proteins in tissues. This approach dramatically enhances the precision of quantitative analyses in immunofluorescence studies, demonstrating IF's capability to provide robust, repeatable quantifications of protein concentrations across a range of biological samples. We have modified the manuscript's Introduction and first Results subsection to discuss the quantitative nature of t-CyCIF.

4. How will the ML/DL trained models here extrapolate to other cancer types in other tissues? Will model training be required for each cancer/tissue type? Would the trained ML/DL model perform similarly on metastatic breast cancer specimens?

- This is a great question but unfortunately not one that we can answer in this manuscript because our datasets are limited to breast cancer. We acknowledge that we do not yet know how these models will perform on other tissue types or cancers, and this will be addressed in future work. For each distinct dataset or cohort, it may be necessary to train a specialized model tailored to the specific data requirements. With methods such as Elastic Net or Light Gradient Boosting Machine (LGBM), a new model must be trained for each protein under investigation. Conversely, an Autoencoder (AE) requires only a single training session, offering a significant advantage by reducing the need for multiple training iterations across different proteins, thus facilitating more efficient model deployment. We have expanded the final paragraphs in the Discussion section to address these limitations more fully and note future opportunities for research.

5. It isn't clear why cytokeratins and hormone receptors exhibited some of the highest variances. The explanation that these proteins are "likely to exhibit patient-specific patterns" is inadequate as it could be argued that this is the case for all stained proteins within the multiplexed imaging dataset.

- We agree that the hypotheses stated in the manuscript about why cytokeratins and hormone receptors exhibited poor imputation performance are not well supported. As a result, we have removed them from the manuscript. Instead, we now write in Results section "Protein abundance imputation with elastic net and light gradient-boosting machines" that (1) the worst performance for imputation can be observed for the highest variance proteins and (2) this observation is not surprising based on general statistics and machine learning theory.

6. While taking the spatial information into consideration is a strength of this work, it isn't clear how neighboring cells were defined herein? How were the different radii sizes related to number of neighboring cells? It also isn't clear why some cells in the sample would not have neighboring cells?

- Thank you. We have expanded the explanation of how neighboring cells are used to quantify spatial information in the Results section "Using cellular spatial information to improve imputation." Briefly,

for each cell within our dataset, we possess spatial coordinates, specifically X and Y values. The resolution of our imaging is such that one pixel equates to 0.65 μm . To analyze cellular interactions and proximities, we have developed an algorithm that identifies each cell, termed the "cell of origin", and aggregates all neighboring cells within a predefined radius, for instance, 15 μm . It is important to note that some cells, particularly immune cells, may exceed the size of the specified radius, leading to instances where certain cells lack neighboring cells. To accommodate this scenario in our analysis, cells without neighbors are assigned a value of zero. The selection of radii for this analysis was informed by a review of extant literature and established norms within the field, ensuring that our methodological choices align with current scientific standards and practices.

7. The summary around RNA probes is not relevant to this work and should be removed in its entirety since only protein staining was evaluated.

- Thank you. We have updated the manuscript to mention RNA probes in only two places that we believe are especially relevant: (1) defining the multiplexed tissue imaging assay and (2) the prior application of imputation in RNA datasets. The first mention of RNA probes in the manuscript helps to clearly define multiplexed tissue imaging. The second mention of RNA probes in the manuscript is important because it highlights important related work. To further address this concern and help prevent confusion about whether our work applies to RNA probes, we have added this sentence in the discussion's limitations: "These analyses demonstrate that it is possible to impute protein abundance in multiplex tissue imaging, but imputation of RNA expression has not been explored."

- We again thank the reviewers for their time and effort in evaluating our manuscript. In this revised version of the manuscript, we made many changes in an effort to address all reviewers' concerns. Below is a point-by-point response to reviewers' concerns as well as a summary of changes made to the manuscript to address these concerns. Reviewers' comments are italicized with wide margins. Our responses are prefixed with an arrow and are indented. **Changes made in the manuscript to address reviewers' comments are highlighted in orange.**

Reviewer #1 (Remarks to the Author):

I thank the authors for addressing many of my concerns, particularly the addition of a patient label prediction task is a good addition. But I maintain that most of the conclusions in the paper are yet supported solely by comparisons of MAE, which is not a biologically meaningful metric. Nor are we given biological intuition for the magnitude of MAE effect sizes, either between model conditions (Null, EN, LGBM, Spatial) or between different markers. This makes the manuscript unidimensional. Other studies cited by this manuscript (scRMD, scImpute) report many dimensions of biologically driven imputation evaluation: MAE yes, but also correlation of RNA concentrations, ARI of cluster results, overlap of differentially expressed genes, cell type identification accuracy, improvement of cell typing, improvement of pathway enrichments etc. The data type of those is transcriptomics, not proteomics, but I believe these are appropriate examples to follow or use as standards.

- Thank you for this expanded dialogue of appropriate evaluation metrics for imputation. We agree that additional metrics are valuable for interpreting our imputation results and their utility. As suggested, we have added many additional metrics to the manuscript:
 - Clustering evaluation comparing the ground truth and imputed protein abundance levels, including the adjusted rand index (ARI) and silhouette score values were added. These demonstrate that imputation improves data and cluster quality for protein expression data. ARI scores range from 0.6 to 0.75, indicating strong cluster agreement. Silhouette scores improved for almost all proteins. The silhouette score decreased for CK19, which also has the lowest ARI score, and these observations correlate with the relatively poor imputation performance of CK19. Together the MAE, ARI scores, and the silhouette score provide evidence that imputed protein expression levels improve data quality for almost all proteins. Figures 2e and 2f have been added to the manuscript to summarize these metrics.
 - Jaccard, adjusted mutual information (AMI), silhouette scores, and ARI from a cell phenotyping task have also been added. For the cell phenotyping task, we (1) used the ground truth data as well as imputed data to call phenotypes. ARI scores are between 0.6 – 0.8 indicating strong to very strong overlap between phenotype using ground truth and imputed data. The Jaccard score

ranges between 0.6 and 0.8 and further supports the hypothesis that imputed data is of similar quality to the ground truth data. These findings are included in Figure 2 and discussed at the end of the section “Protein abundance imputation with elastic net and light gradient-boosting machines”. We added an additional supplemental figure S4, to show the AMI and silhouette scores for phenotype calling as well. AMI scores are also in the 0.6-0.8 range, providing additional evidence that the imputed data preserves biological information. The silhouette scores for the ground truth data are consistently between 0.25 and 0.3, and silhouette scores using imputed data are between 0.20 and 0.28. These results indicate that cluster compactness is reduced using imputed data but is still close to the ground truth data. The notable exception is CK19, where silhouette scores are much lower.

Together, these additional metrics provide strong evidence that (1) the imputed data is quantitatively very similar to the ground truth data and (2) that there is strong biological meaning in the imputed data as evidenced by the metrics computed from the cell phenotyping task.

One easy dimension to add, asked for in review 1 but not done in the revision, would be in situ visualizations of imputed vs ground truth expression in different model contexts, which adds biological intuition to quantitative differences.

- We apologize for not addressing this request in the first revision of the manuscript. We have revised Figure 2 to include in situ images alongside the ground truth and imputed values. It is important to note that while the in situ images display pixel intensities, the ground truth and imputed data are single-cell intensities obtained from image processing. Therefore, quantitative differences cannot be directly observed when comparing in situ images to the ground truth single-cell data from image processing. Nonetheless, these side-by-side visualizations enhance the visual context of the imputed data and make it possible to compare imputed data to the original image and ground truth data.

Cell-wise correlations would likewise give a sense of single cell imputation accuracy and the distribution of marker expression across single cells that is simply not captured by error level.

- We apologize for not addressing this request in the first revision of the manuscript. We have added a correlation graph to the supplemental figures which show that there is moderate to strong correlation between imputed and ground truth data. Correlation is between 0.4 (CK19) and 0.8 (EGFR). The observed correlation values are within the typical range observed for scRNA imputation tools, such as DRImpute, MAGIC, and ScImpute, which yield cell-to-cell correlation values between 0.4 and 0.8 (Xu et al., 2020). The correlation combined with the ARI and

the silhouette scores give an insight into the models ability to improve cluster quality while still preserving the underlying biological information. ARI scores for protein expression values are between 0.6 and 0.75, indicating a strong overlap between ground truth and imputed data further supporting our claim that the imputation does work.

The addition of the tile-treatment predictions is a step in the right direction, but still a bit messy. The authors used the LGBM model because it was most accurate, but not the spatial LGBM model, which was actually most accurate according to the preceding conclusions. Why?

- This is a great question. A simple LGBM model that did not incorporate spatial features was used because the spatial LGBM model requires significant computational resources to train and evaluate. We estimate that using a spatial LGBM model would require ~1 month of computational time for this analysis because of the need to regenerate all necessary features for both imputation and the downstream task. However, in the manuscript we have demonstrated success in the downstream task using the simple LGBM model without spatial features. Given this success, we anticipate that the spatial LGBM model would yield equal or better performance to the simple LGBM model because the spatial model was demonstrated to be modestly more accurate than the simple LGBM model in the primary task of protein imputation. In summary, the use of the simple LGBM model is sufficient to demonstrate success on the downstream task and hence the utility of the imputed data.

Their result required a fairly involved preprocessing pipeline that includes tile pre-selection. By itself that is OK. But the actual result raises questions as well. First, what exactly is presented? Is it: For each marker, when imputing that marker, the error of predicting treatment status...? Perhaps the Y axis label is incorrect, because increased error would mean worse performance, no? And MAEs close to 1 would be very high in this context?

- The title of figure 7 was incorrect, and we apologize for the confusion that this caused. Figure 7 shows the accuracy of the trained model on the single cell timepoint prediction task. Zero represents all incorrect predictions from the model, and 1 denotes all correct predictions. Thus, our results in Figure 7 demonstrate significant accuracy improvement in the prediction task using imputed data. We hypothesize that this is due to the imputed data removing noise from the ground truth data. With less noise, the imputed data led to better performance on the prediction task. Reducing noise not only enhances data quality but also complements the effects of upsampling, which can amplify meaningful patterns in the data. The established benefits of upscaling techniques further support our hypothesis, as demonstrated in (Gondara & Wang, 2018; Patruno et al., 2020; Tran et al., 2022).

Second, unless I'm mistaken, the tile selection itself uses all the markers, including those markers that will later be imputed, which is an unrealistic scenario.

- Thank you for this observation. We have removed the imputed marker from the tile selection process, performed the experiment with the imputed data, and recreated Figure 7 using the imputed data. With this methodology, prediction models using the imputed data continue to perform at the same level or better than models using the ground truth data. These results demonstrate that the imputed data can be used in a biologically meaningful task.

Third, it seems to me that a successful experiment would mean that, in the absence of Marker X, sample level predictions suffer, then predictions can be restored to some extent using imputation. That would be a very strong result.

- We observe this result for several proteins, including ER, CK19, and CD45. When imputed data is used for these proteins, prediction accuracy does indeed suffer compared to accuracy with ground truth data. However, as discussed above, we (1) do not observe this result for all proteins and (2) hypothesize that noise in the ground truth data reduces prediction accuracy whereas imputed data does not suffer from this noise and can be used to make better predictions.

Perhaps the authors could just do tile classification as in the first step of the current pipeline, but do those with ground truth data, missing data, and imputed data? That would be more interpretable than the cell level classing shown in Fig 7d.

- We appreciate the suggestion to make predictions at the tile level. However, there are several studies that demonstrate the value and accuracy of single-cell predictions (Cannoodt et al., 2021; De Biasi et al., 2024; Ren et al., 2023). These studies inspired our single-cell prediction task. We also believe the single-cell prediction task is appropriate given the single-cell imputation performed in the manuscript. Finally, performance on the single-cell task is quite good and demonstrates the value of the imputed data on a meaningful biological task. For these reasons, we prefer to report single-cell predictions rather than tile-level predictions. We hope our reasoning for choosing a single-cell prediction task rather than a tile-level prediction task is appreciated but welcome additional conversation if needed.

The authors have done well to add treatment-related predictions, some control experiments, and a generalization figure (the second CyCIF dataset). I hope I am being consistent with my ground truth review in saying that it still lacks:

- 1. Multidimensional evaluation of imputation success*
- 2. A clean experiment showing the impact of imputation on treatment predictions.*

- Thank you for this summary. We have provided a point-by-point response to these requests above.

We sincerely thank the reviewers for their time and thoughtful evaluation of our manuscript. In this revised version, we have addressed all minor points raised during the review process. Below, we provide a point-by-point response to each reviewer comment, along with a summary of the corresponding revisions made to the manuscript. Reviewer comments are italicized and presented in wide margins; our responses are indented and prefixed with an arrow for clarity.

Reviewer #1 (Remarks to the Author):

I think the authors have addressed my concerns satisfactorily and written good clarifications into the text. Thanks for your replies. I feel a little 'weird' about imputed predictions outperforming ground truth in the prediction task, but accept the rationale of de-noising as plausible.

The following are only minor points to improve readability or clarity, if the authors or editor should choose:

** Adjust contrast or thresholds of the in situ / cell expression level images (2c) to make it more visible.*

- Thank you for your recommendation. We agree and have adjusted the brightness of panel 2c to enhance its visibility and overall clarity.

** Why only 5 proteins for additional metrics (e.g. Jaccard)? Show all, or explain why just a subset.*

- Thank you for your question. Only five proteins contribute to phenotype calling and are therefore relevant to the experiment. We have added clarifying text to the manuscript to highlight this point.

** Addition of colors with no legend (Fig. 2e, g, h) is slightly confusing.*

- Thank you for bringing this to our attention. The referenced subpanels have now been incorporated into Figure 3. We have removed the coloring from panels a and b, as suggested. We retained the colors in panels c and d, as they highlight the same proteins used for phenotype calling and aid in visual interpretation.

** In review doc authors wrote: "the spatial LGBM model requires significant computational resources to train and evaluate." Maybe add this justification of model choice to the paper, or at least mention it when spatial LGBM performs best.*

- Thank you for this recommendation. We have added explanatory text to the manuscript clarifying our rationale for using the LGBM model instead of the spatial LGBM, highlighting the specific considerations that informed this choice.

** The large point size in 7b inset makes the plot confusing, gives the impression points show something more than cell locations.*

- Thank you for your feedback. We have reduced the point size in the relevant figure to improve clarity and minimize potential confusion, as suggested.

Reviewer #1 (Remarks on code availability):

Nice readme!